

# Quantifying the probability and uncertainty of multiple-structure rupture and recurrence intervals in Taiwan

Chieh-Chen Chang[1], Chih-Yu Chang[1], Chung-Han Chan[1,2]

[1]Department of Earth Sciences, National Central University, Taoyuan, 32001, Taiwan

[2]Earthquake-Disaster & Risk Evaluation and Management (E-DREaM) Center, National Central University, Taoyuan, 32001, Taiwan

*Correspondence to*: Chung-Han Chan (hantijun@googlemail.com)

**Abstract.** This study identifies structure pairs with potential for simultaneous rupture in a coseismic period via Coulomb stress change, and quantifies their rupture recurrence intervals and uncertainties according to the Gutenberg-Richter law and

the empirical formula of rupture parameters. To assess the potential for a multiple-structure rupture, we calculated the probability of Coulomb stress triggering between seismogenic structures. We assumed that a multiple-structure rupture would occur if two structures could trigger each other by enhancing the plane with thresholds of a Coulomb stress increase and the distance between the structures. According to different thresholds, we identified various sets of seismogenic structure pairs. To estimate the recurrence intervals for multiple-structure ruptures, we implemented a scaling law and the Gutenberg-

Richter law in which the slip rate could be partitioned based on the magnitudes of the individual structure and multiple-structure ruptures. In addition, considering that a single structure may be involved in multiple cases of multiple-structure ruptures, we developed new formulas for slip partitioning in a complex fault system. By implementing the range of slip area and slip rate of each structure, the magnitudes and recurrence intervals of multiple-structure ruptures could be estimated. Due to a larger characteristic magnitude and a larger displacement of the multiple-structure rupture, its recurrence interval

could be longer. Therefore, application of the multiple-structure rupture could lead to an increase in seismic hazard in a long return period, which would be crucial for the safety evaluation of infrastructures, such as nuclear power plants and dams.

## 1 Introduction

A rupture taking place along several fault segments or/and structures can cause an earthquake with a larger magnitude (Wells and Coppersmith, 1994) and often leads to disaster. The 1935 $M_L$7.1 Hsinchu-Taichung, Taiwan, earthquake is an example.

This event is attributed to a rupture on the Shihtan and Tunzijiao faults and resulted in more than 3,000 fatalities and the destruction of more than 60,000 buildings. According to the earthquake parameter scaling relationship (Wells and Coppersmith, 1994), either the Shihtan or Tunzijiao fault could cause an earthquake with a maximum magnitude of only 6.6 (Wang et al., 2016). This case raises the importance of multiple-structure ruptures on seismic hazard assessment.

Thus, the Taiwan Earthquake Model (TEM) has considered the possibility of several multiple-structure ruptures by a

probabilistic seismic hazard assessment for Taiwan (Chan et al., 2020). Their model implemented a seismogenic structure





database summarized by Shyu et al. (2020) that identified possible multiple-structure ruptures based on geomorphological and geological evidence. To quantify their recurrence intervals, Chan et al. (2020) proposed a procedure for partitioning the slip rate of each individual structure to multiple structures. In their procedure, the case that one structure could be associated with multiple pairs was not specified.

Thus, this study aims to identify structures that could rupture simultaneously based on a physics-based model and propose a set of formulas to evaluate their recurrence intervals. The possibility of a multiple-structure rupture is determined based on the Coulomb stress change imparted by each structure and the distance from one to the other. Quantifying the recurrence interval relies on a scaling law (Wells and Coppersmith, 1994) and the Gutenberg-Richter law (Gutenberg and Richter, 1944). Our approach's procedure is transparent and can be applied to reexamining the composite ruptures of the seismogenic

structure system in Taiwan and other regions, which is beneficial to subsequent probabilistic seismic hazard assessment.

## 2 Distinguishing possible seismogenic structure pairs according to Coulomb stress change

Previous studies (e.g., Catalli and Chan, 2012) have concluded that changes in the Coulomb stress resulting from previous earthquakes could trigger the occurrence of subsequent events in adjacent areas. Such an approach would be especially applicable to determining the interaction between two fault systems if their rupture mechanisms are well-understood. In the

following, we introduce the Coulomb failure criterion to discuss interaction between structure systems, then distinguish seismogenic structure pairs that could rupture simultaneously in a coseismic period, considering different criteria.

### 2.1 Introduction of Coulomb stress

The Coulomb failure criterion describes mainly the characteristics of material failure (King et al., 1994; Toda et al., 1998). The criterion illustrates a plane encountering stress change, which could be decomposed into two vectors, shear stress

change, $\Delta \tau$, and normal stress, $\Delta \sigma_n$:

$$\Delta CFS = \Delta \tau - \mu' \Delta \sigma_n ,$$   (1)

where $\Delta CFS$ is the Coulomb stress change, and $\mu'$ is the effective friction coefficient. In this study, we used the COULOMB 3.4 software (Toda et al., 2011) for calculation of Coulomb stress change.

The theory mentioned above could quantify the Coulomb stress change imparted by a rupture on a fault (the "source fault")

on another fault that receives the stress (the "receiver fault"). Assuming the source fault possesses a reverse mechanism (i.e., rake angle of 90˚, as shown in Fig. 1a), the rupture on the source fault could result in a Coulomb stress decrease if there is another reverse fault system on the hanging wall or the footwall, whereas faults located at the extensions along both sides of the source fault plane could be triggered. This theory is also applicable to a case where the source fault possesses different





mechanisms from the receiver fault. A fault plane with a reverse mechanism could enhance the rupture on a right-lateral fault system at the tip of the strike orientation on the hanging wall (Fig. 1b).

Based on the Coulomb stress change, we could quantify the possibility of a coseismic rupture for two fault systems. To explore the interactions between seismogenic structures in Taiwan, detailed structural parameters should be considered.

## 2.2 Possible coseismic multiple-structure rupture defined by the Coulomb stress transfer

To understand stress interaction between seismogenic structures in Taiwan, we accessed the TEM database, which incorporates 45 seismogenic structures (Shyu et al., 2016; 2020, structure alignment shown in Fig. 2) and corresponding parameters (shown in Table 1). According to the surface trace and dipping angles, the three-dimensional geometry of each structure is illustrated by pieces of sub-faults.

Since these structures could initiate earthquakes and trigger neighboring structures, we investigated their potential interaction through Coulomb stress change. We considered a characteristic earthquake with corresponding slip (shown in Table 1) on each structure and evaluated the Coulomb stress change solved on each sub-fault of the other structures. Previous studies concluded that stress increases greater than a threshold could trigger subsequent earthquakes. For example, Ma et al. (2005) and Stein (2004) suggested that stress increases greater than 0.1 and 0.01 bar, respectively, could trigger seismicity activity. Assuming that a structure could be triggered if half its plane was enhanced with a stress increase greater than a threshold, we identified potential structural pairs that could trigger each other, considering different stress thresholds, and discuss their credibility. Close distance between two structures is another key factor of rupture triggering. For example, the UCERF3 (Uniform California Earthquake Rupture Forecast, Version 3; Field et al., 2015) defines two faults that could rupture simultaneously if the distance between the two is less than 5 km. We first follow this criterion to identify paired structures and then discuss their impact when different distance thresholds were assumed.

Following the assumptions mentioned above, we could identify seismogenic structure pairs that could rupture in a coseismic period. We first considered the stress threshold of $\Delta$CFS $\geq$ 0.1 and distance threshold of 5 km to identify potential rupture pairs. For example, in the relation between the Meishan fault (ID 20) and the Chiayi frontal structure (ID 21), if the rupture initiates on the Chiayi frontal structure, the stress on the Meishan fault plane would be disturbed significantly. In that instance, 72% of the fault plane could be enhanced by more than 0.1 bar of the Coulomb stress. Such interaction could also be confirmed by a simplified Coulomb stress model (Fig. 1b), in that a reverse-mechanism fault's ruptures (such as the Chiayi frontal structure) would trigger a fault with a right-lateral mechanism to the northeast (such as the Meishan fault) closer to failure. On the other hand, a rupture on the Meishan fault (ID 20) could result in 64% of the Chiayi frontal structure (ID 21) plane experiencing a stress increase of more than 0.1 bar. Our results show that either of the two seismogenic structures could trigger more than 50% of the other structure plane. In addition, based on the three-dimensional geometries of the two seismogenic structures, their closest distance is 1.87 km, which meets our proximity criteria (< 5 km). Therefore, we conclude that the Meishan fault and the Chiayi frontal structure can mutually induce a coseismic rupture.



According to the ratio by which each structure plane is triggered by other structures (Table S1) and the distance between each of two structures (Table S2), we defined 17 pairs of seismogenic structures that could potentially rupture in a coseismic period (Table 2).

We further identified potential multiple-structure pairs through different thresholds of stress changes and distances.
Considering $\Delta CFS$ of 0.01 bar as a lower bound of stress triggering (Stein, 2004), we proposed four sets of stress increase thresholds (0.01, 0.05, 0.1, and 0.2 bars, respectively), as well as two threshold sets for the distance between structures (2.5 and 5.0 km, respectively). Based on the criteria, multiple-structure pairs were identified (Table 3). More structure pairs were expected if a lower $\Delta CFS$ threshold and/or a longer maximum distance were assumed and vice versa. The number of identified pairs is between 6 ($\Delta CFS \geq 0.2$ bar, distance $\leq 2.5$ km) and 34 ($\Delta CFS \geq 0.01$ bar, distance $\leq 5.0$ km).
We have identified potential structures that might rupture in a coseismic period. To understand the activities of these multiple-structure rupture cases, next we will propose a procedure to evaluate their recurrence intervals.

## 3 Recurrence interval of the multiple-structure rupture

The recurrence interval is a critical parameter in probabilistic seismic hazard analysis. Here, we are going to calculate the recurrence interval of multiple-structure ruptures and discuss their impact on seismic hazards.

### 3.1 Recurrence interval of multiple-structure ruptures

The rupture recurrence interval (denoted as $R_{L1}$) of a single seismogenic structure ($L1$), $R_{L1}$, can be evaluated as the ratio of slip of a characteristic earthquake and slip rate (denoted as $D_{L1}$ and $\dot{D}_{L1}$, respectively):

$$R_{L1} = \frac{D_{L1}}{\dot{D}_{L1}}. \tag{2}$$

To evaluate the seismic rate of a multiple-structure rupture on two seismogenic structures ($L1$ and $L2$), we implemented the
Gutenberg-Richter law to describe the relationship between earthquake frequency $N$ and magnitude $M$:

$$log(N) = a - bM. \tag{3}$$

Considering the different moment magnitudes between single-structure and multiple-structure ruptures, the ratio of earthquake frequency and slip-rate partitioning could be evaluated. The moment magnitude ($M_w$) of the multiple-structure rupture could be evaluated according to the rupture area (denoted as $A$) and fault types of the two seismogenic structures:

$M_w = 4.33 + 0.90 \times log(A)$ ... for reverse faulting; $\qquad (4)$

$M_w = 3.98 + 1.02 \times log(A)$ ... for strike-slip faulting; $\qquad (5)$

$M_w = 3.93 + 1.02 \times log(A)$ ... for normal faulting. $\qquad (6)$




Based on the scale ($M_w = \frac{\log M_0}{1.5} - 10.73$, derived by Kanamori, 1977) and the definition of seismic moment ($M_0 = \mu A D$, $\mu = 3 \times 10^{10} \, N/m^2$), average displacement of a seismogenic structure ($D$, in meters) could be evaluated according to $M_w$ and $A$ (in km²):

$$D = \frac{10^{(M_w+10.73)\times\frac{2}{3}} \times 10^{-12}}{3 \times 10^{11} \times A}. \tag{7}$$

The potential of multiple-structure ruptures could be attributed to the moment accumulation from the first and second structures, $L1$ and $L2$. We assumed their slip rates, $\dot{D}_{L1}$ and $\dot{D}_{L2}$, could be partitioned into two cases, the rupture on the original structure and the rupture on multiple structures. The slip rate partitioned to the multiple-structure rupture from $L1$ and $L2$ can be represented as:

$$\dot{D}_{L1+L2}^{L1} = C_1 \times \dot{D}_{L1}' \text{ and} \tag{8}$$

$$\dot{D}_{L1+L2}^{L2} = C_2 \times \dot{D}_{L2}', \text{ respectively,} \tag{9}$$

where $\dot{D}_{L1}$ and $\dot{D}_{L2}$ represent the original slip rates of $L1$ and $L2$, respectively, and $C_1$ and $C_2$ represent the obtained partitioned rates from $L1$ and $L2$, respectively. Assuming the seismicity ratio between $L1$, $L2$ and $L1+L2$ is based on their magnitudes and the Gutenberg-Richter law (shown in equation 1), then:

$$C_1 = \frac{10^{b(M_{L1}-M_{L1+L2})} \times D_{L1+L2}}{D_{L1}} \tag{10}$$

and

$$C_2 = \frac{10^{b(M_{L2}-M_{L1+L2})} \times D_{L1+L2}}{D_{L2}}, \tag{11}$$

where $M_{L1}$ and $M_{L2}$ represent the magnitudes of $L1$ and $L2$, respectively; $D_{L1}$ and $D_{L2}$ represent the displacements of $L1$ and $L2$, respectively; $M_{L1+L2}$ represents the magnitude of the multiple-structure rupture; and $D_{L1+L2}$ represents the displacement of the multiple-structure rupture.

According to the obtained partitioned rates, the slip rate partitioned to individual structure ruptures ($L1$ and $L2$, respectively) can be represented as:

$$\dot{D}_{L1}' = \frac{\dot{D}_{L1}}{(\frac{A_{L1+L2}}{A_{L1}} \times C_1 + 1)} \qquad \text{and} \tag{12}$$

$$\dot{D}_{L2}' = \frac{\dot{D}_{L2}}{(\frac{A_{L1+L2}}{A_{L2}} \times C_2 + 1)}, \text{ respectively.} \tag{13}$$



where $A_{L1}$ and $A_{L2}$ represent the rupture areas of $L1$ and $L2$, respectively, and $A_{L1+L2}$ represents the area of the multiple-structure rupture.

By integrating the obtained partitioned rates (equations 10 and 11) and the slip rate partitioned to individual structure
ruptures (equations 12 and 13), the slip rate partitioned to the multiple-structure rupture from the original structures can be obtained (equations 8 and 9). Then the sum of the slip rates for the multiple-structure rupture is calculated using the partitioned rates of the two structures, represented as:

$$\dot{D}_{L1+L2} = \dot{D}^{L1}_{L1+L2} + \dot{D}^{L2}_{L1+L2}. \tag{14}$$

Considering the displacement and slip rate, recurrence intervals for individual structures ($R_{L1}$ and $R_{L2}$) and the multiple-
structure rupture ($R_{L1+L2}$) can be represented as:

$$R_{L1} = \frac{D_{L1}}{\dot{D}_{L1}'}, \tag{15}$$

$$R_{L2} = \frac{D_{L2}}{\dot{D}_{L2}'}, \text{ and} \tag{16}$$

$$R_{L1+L2} = \frac{D_{L1+L2}}{\dot{D}_{L1+L2}}, \text{ respectively.} \tag{17}$$

Here we take the case of the Hsinhua fault (ID 24, denoted as $L_1$) and the Houchiali fault (ID 25, denoted as $L_2$) as an
example. We followed the results of Wang et al. (2016) and assumed a fixed b-value of 1.10. The fault types of Hsinhua fault and the Houchiali fault structures are the strike-slip fault and reverse fault, respectively, and their $M_w$ are 6.38 and 6.07, respectively (Table 1). Since the characteristic earthquake of the Hsinhua fault is larger, we assume a strike-slip mechanism for the multiple-structure rupture with rupture area, magnitude, and displacement as the following:

$$A_{L1+L2} = 222.89 + 86.25 = 309.14 \; km^2;$$

$$M_{w\,L1+L2} = 3.98 + 1.02 \times log\,(309.14) = 6.52;$$

$$D_{L1+L2} = \frac{10^{(6.52+10.73)\times\frac{2}{3}} \times 10^{-12}}{3\times10^{11}\times309.14} = 0.809 \; m.$$

The slip-partitioned ratios from the Hsinhua fault and the Houchiali fault were:

$$C_1 = \frac{10^{1.1\times(6.38-6.52)}\times0.809}{0.69} = 0.82 \text{ (Hsinhua fault), and}$$

$$C_2 = \frac{10^{1.1\times(6.08-6.52)}\times0.809}{0.61} = 0.42 \text{ (Houchiali fault).}$$

The slip rates partitioned to the original structures were:

$$\dot{D}_{L1}' = \frac{2.65}{(\frac{309.14}{222.89}\times0.82+1)} = 1.24 \; mm/year \text{ (Hsinhua fault), and}$$





$$\dot{D}_{L2}' = \frac{7.07}{\left(\frac{309.14}{86.25} \times 0.42 + 1\right)} = 2.822 \; mm/year \; \text{(Houchiali fault)}.$$

The slip rates partitioned to the multiple-structure rupture were:

$$\dot{D}_{L1+L2}^{L1} = 0.82 \times 1.24 = 1.017 \; mm/year \; \text{(Hsinhua fault), and}$$

$$\dot{D}_{L1+L2}^{L2} = 0.42 \times 2.822 = 1.185 \; mm/year \; \text{(Houchiali fault)}.$$

The sum of the slip rates for the multiple-structure rupture was:

$$\dot{D}_{L1+L2} = 1.017 + 1.185 \; = 2.202 \; mm/year.$$

The recurrence intervals for each individual structure and the multiple-structure rupture were:

$$R_{L1} = \frac{0.69}{1.24} \times 1000 = 556 \; years \; \text{(Hsinhua fault)},$$

$$R_{L2} = \frac{0.61}{2.822} \times 1000 = 216 \; years \; \text{(Houchiali fault), and}$$

$$R_{L1+L2} = \frac{0.808}{2.202} \times 1000 = 367 \; years \; \text{(multiple-structure rupture)}.$$

**3.2 Single structure contributes to several multiple-structure ruptures**

A single seismogenic structure could be involved in multiple cases of multiple-structure rupture. For such cases, however, evaluation of the corresponding recurrence intervals has seldom been discussed. Here, we propose a procedure for 180 quantifying the return period of this case, shown below.

When a single structure ($L_1$) is involved in multiple cases of multiple-structure rupture ($L1+L2, \ldots, L1+Ln$), the slip rate partitioned to the original structure can be obtained based on the revision of equation (10), represented as:

$$\dot{D}_{L1}' = \frac{A_{L1} \times \dot{D}_{L1} \times D_{L1}}{(A_{L1} \times D_{L1}) + \sum_{i=2}^{n}(A_{L1+Li} \times D_{L1+Li} \times 10^{b(M_{L1}-M_{L1+Li})}) + \sum_{i=2}^{n-1}\sum_{j=3}^{n}(A_{L1+Li+Lj} \times D_{L1+Li+Lj} \times 10^{b(M_{L1}-M_{L1+Li+Lj})}) + \sum_{i=2}^{n-2}\sum_{j=3}^{n-1}\sum_{k=4}^{n} \cdots}, \; 1 < i <$$

j < k. (18)

where $D_{L1+L2}, \ldots, D_{L1+Ln}$ represent the displacements of the multiple-structure rupture cases $L1+L2, \ldots, L1+Ln$, respectively.

The slip rate partitioned to the multiple-structure rupture cases $L1+L2, \ldots, L1+Ln$ can be represented as:

$$\dot{D}_{Lx}^{L1} = \frac{A_{L1} \times \dot{D}_{L1} \times D_{L1+Lx} \times 10^{b(M_{L1}-M_{Lx})}}{(A_{L1} \times D_{L1}) + \sum_{i=2}^{n}(A_{L1+Li} \times D_{L1+Li} \times 10^{b(M_{L1}-M_{L1+Li})}) + \sum_{i=2}^{n-1}\sum_{j=3}^{n}(A_{L1+Li+Lj} \times D_{L1+Li+Lj} \times 10^{b(M_{L1}-M_{L1+Li+Lj})}) + \sum_{i=2}^{n-2}\sum_{j=3}^{n-1}\sum_{k=4}^{n} \cdots}, \; Lx =$$

$L1 + Li + Lj + Lk + \cdots$ (19)





respectively. In this case, evaluation of the recurrence interval for each multiple-structure rupture requires the slip rates

contributed from two structures as well, similar to what is shown in equation (14). The total slip rate of each case of multiple-structure rupture can be represented as:

$$\dot{D}_{Lx} = \dot{D}_{Lx}^{L1} + \sum_{i=2}^{n} \dot{D}_{Lx}^{Li}. \tag{20}$$

The recurrence intervals for the original structure and each multiple-structure rupture case can be represented as:

$$R_{L1} = \frac{D_{L1}}{\dot{D}_{L1}\prime}, \tag{21}$$

and

$$R_{Lx} = \frac{D_{Lx}}{\dot{D}_{Lx}}, \text{ respectively.} \tag{22}$$

Here we took the cases of the Chiayi frontal structure (ID 21, here denoted as $L1$) with the Meishan fault (ID 20, here denoted as $L2$) and the Tainan frontal structure (ID 41, here denoted as $L3$) as an example. The Chiayi frontal structure is reverse faulting with the potential for an $M_w$7.21 earthquake; the Meishan fault is strike-slip faulting with the potential for an

$M_w$6.60 earthquake; the Tainan frontal structure is reverse faulting with the potential for an $M_w$7.24 earthquake. Considering the magnitude of each structure, we assumed these two pairs of multiple structures are reverse faulting, and evaluated their fault areas and moment magnitudes accordingly, as follows:

$A_{L1+L2} = 371.7 + 1580.88 = 1952.58 \ km^2$ (area for ID20 + ID21);

$A_{L1+L3} = 1580.88 + 1722.64 = 3303.52 \ km^2$ (area for ID21 + ID41);

$M_{w\,L1+L2} = 4.33 + 0.90 \times log\,(1952.58) = 7.29$ (magnitude for ID20 + ID21);

$M_{w\,L1+L3} = 4.33 + 0.90 \times log\,(3303.52) = 7.5$ (magnitude for ID21 + ID41);

$$D_{L1+L2} = \frac{10^{(7.29+10.73)\times\frac{2}{3}}\times 10^{-12}}{3\times10^{11}\times1952.58} = 1.829 \ m \text{ (displacement for ID20 + ID21);}$$

$$D_{L1+L3} = \frac{10^{(7.5+10.73)\times\frac{2}{3}}\times 10^{-12}}{3\times10^{11}\times3303.52} = 2.233 \ m \text{ (displacement for ID21 + ID41);}$$

$$\dot{D}_{L1}\prime = \frac{1580.88\times3.36\times1.71}{(1580.88\times1.71)+(1952.58\times1.829\times10^{1.1\times(7.21-7.29)})+(3303.52\times2.233\times10^{1.1\times(7.21-7.5)})} = 0.992 \ mm/year;$$

$$\dot{D}_{L1+L2}^{L1} = \frac{1580.88\times3.36\times1.84\times10^{1.1\times(7.21-7.29)}}{(1580.88\times1.71)+(1952.58\times1.829\times10^{1.1\times(7.21-7.29)})+(3303.52\times2.233\times10^{1.1\times(7.21-7.5)})} = 0.866 \ mm/year; \text{ and}$$

$$\dot{D}_{L1+L3}^{L1} = \frac{1580.88\times3.36\times2.21\times10^{1.1\times(7.21-7.5)}}{(1580.88\times1.71)+(1952.58\times1.829\times10^{1.1\times(7.21-7.29)})+(3303.52\times2.233\times10^{1.1\times(7.21-7.5)})} = 0.621 \ mm/year.$$




Since the Meishan fault and the Tainan frontal structure contributed to only one multiple-structure rupture pair, respectively, $\dot{D}_{L1+L2}^{L2}$ and $\dot{D}_{L1+L3}^{L3}$ can be calculated according to equation (13).

$\dot{D}_{L1+L2}^{L2} = 0.312 \; mm/year; \dot{D}_{L1+L3}^{L3} = 0.268 \; mm/year;$

$\dot{D}_{L1+L2} = 0.866 + 0.312 = 1.178 \; mm/year;$

$\dot{D}_{L1+L3} = 0.621 + 0.268 = 0.889 \; mm/year;$

$R_{L1} = \frac{1.71}{0.992} \times 1000 = 1724 \; years$ (recurrence interval of ID21);

$R_{L1+L2} = \frac{1.829}{1.178} \times 1000 = 1553 \; years$ (recurrence interval of ID20 + ID21); and

$R_{L1+L3} = \frac{2.233}{0.889} \times 1000 = 2512 \; years$ (recurrence interval of ID21 + ID41).

A single earthquake could be attributed to multiple (more than three) structures (for example, the 2016 $M_w7.8$ Kaikōura, New Zealand, earthquake; see Shi et al., 2017, in detail). In such special cases, the recurrence interval can be also evaluated through the procedure mentioned above. For example, the Chiayi frontal structure (ID 21, here denoted as $L1$) could trigger the Meishan fault (ID 20, here denoted as $L2$) and the Tainan frontal structure (ID 41, here denoted as $L3$), respectively, in some criteria (Table 3), inferring the possibility of multiple ruptures in an event. We assumed this event is reverse faulting 225 and evaluated its fault area and moment magnitude accordingly, described in the following:

$A_{L1+L2+L3} = 371.7 + 1580.88 + 1722.64 = 3675.22 \; km^2;$

$M_{w L1+L2+L3} = 4.33 + 0.90 \times log(3675.22) = 7.54;$

$D_{L1+L2+L3} = \frac{10^{(7.54+10.73) \times \frac{2}{3}} \times 10^{-12}}{3 \times 10^{11} \times 3675.22} = 2.305 \; m;$

$\dot{D}_{L1}' = \frac{1580.88 \times 3.36 \times 1.71}{(1580.88 \times 1.71)+(1952.58 \times 1.829 \times 10^{1.1 \times (7.21-7.29)})+(3303.52 \times 2.233 \times 10^{1.1 \times (7.21-7.5)})+(3675.22 \times 2.305 \times 10^{1.1 \times (7.21-7.54)})} = 0.708 \; mm/year;$

$\dot{D}_{L1+L2+L3}^{L1} = \frac{1580.88 \times 3.36 \times 2.305 \times 10^{1.1 \times (7.21-7.54)}}{(1580.88 \times 1.71)+(1952.58 \times 1.829 \times 10^{1.1 \times (7.21-7.29)})+(3303.52 \times 2.233 \times 10^{1.1 \times (7.21-7.5)})+(3675.22 \times 2.305 \times 10^{1.1 \times (7.21-7.54)})} = 0.414 \; mm/year;$

$\dot{D}_{L1+L2+L3}^{L2} = \frac{371.7 \times 2.51 \times 2.305 \times 10^{1.1 \times (6.6-7.54)}}{(371.7 \times 0.89)+(1952.58 \times 1.829 \times 10^{1.1 \times (6.6-7.29)})+(3675.22 \times 2.305 \times 10^{1.1 \times (6.6-7.54)})} = 0.114 mm/year;$ and

$\dot{D}_{L1+L2+L3}^{L3} = \frac{1722.64 \times 0.92 \times 2.305 \times 10^{1.1 \times (7.24-7.54)}}{(1722.64 \times 1.74)+(3303.52 \times 2.233 \times 10^{1.1 \times (7.24-7.5)})+(3675.22 \times 2.305 \times 10^{1.1 \times (7.24-7.54)})} = 0.159 mm/year.$

Note that L2 and L3 will not rupture together:

$\dot{D}_{L1+L2+L3} = 0.414 + 0.114 + 0.159 = 0.687 \; mm/year;$

$R_{L1} = \frac{1.71}{0.708} \times 1000 = 2415 \; years;$ then





$R_{L1+L2+L3} = \frac{2.305}{0.687} \times 1000 = 3355\ years.$

### 3.3 The results of multiple-structure rupture recurrence intervals

According to the structure parameters (Table 1), the recurrence intervals of each pair of potential multiple-structure ruptures can be evaluated (Table 2). Here, we consider the 17 pairs with ΔCFS ≥ 0.1 bar and distance ≤ 5.0 km and evaluated their

potential magnitudes and recurrence intervals by implementing the range of slip area and slip rate of each structure (Table 1). The largest magnitude is expected if the maximum slip areas of the two structures are assumed (based on equations 4-6). Also, the shortest recurrence interval is expected if the minimum slip area and maximum slip rate are assumed (based on equations 4-17).

In comparison with the recurrence intervals of the original structures without considering a multiple-structure rupture (Table

1), longer recurrence intervals are expected for multiple-structure ruptures and individual structures due to slip partitioning. For example, the recurrence interval of the Chiayi frontal structure (ID 21) has been extended from 510 to 1,724 years. Based on these results, the seismic hazard level for a short return period (e.g., 475 years, corresponding to a 10% probability in 50 years) would be lower.

Additionally, our results show that a single seismogenic structure sometimes pairs with several cases of multiple-structure

ruptures. For example, the Hukou fault (ID 4) potentially ruptures with the Shuanglianpo structure (ID 2), the Fengshan river strike-slip structure (ID 5), and the Hsinchu fault (ID 6), while the Hsinchu fault (ID 6) could also result in multiple-segment ruptures with the Hsinchu frontal structure (ID 8) and the Touhuanping structure (ID 9). Besides these two cases associated with three rupture pairs, several structures could be associated with two multiple-structure pairs (Table 2), raising the importance of implementing slip partitioning from a single structure to several multiple-structure ruptures. Based on our

analysis, the structures that pair with several cases of multiple-structure ruptures might be difficult to rupture solely. That is, based on equations 18 and 21, the slip rate of these structures could be partitioned to several cases of multiple-structure ruptures, resulting in longer recurrence intervals. For example, the Hukou fault (ID 4) and the Hsinchu fault (ID 6) involved four and three pairs of multiple-structure ruptures, respectively (Table 2), and their recurrence intervals become 4.4 and 5.3, respectively, longer than the cases without considering multiple-structure ruptures (Table 4).

## 4 Discussion and conclusion

### 4.1 Interaction between structures and possible coseismic ruptures

In this study, we explored possible coseismic multiple-structure ruptures and quantified their recurrence intervals by implementing the Coulomb stress change and the Gutenberg-Richter law, respectively. The analyzing procedure we proposed is based on physics- and statistics-based models, and the outcomes are reproducible.





We compared our results with the conclusion of Shyu et al. (2020) that some seismogenic structure pairs could rupture simultaneously, such as the Hsinchu fault (ID 6) and the Hsinchu frontal structure (ID 8), the Touhuanping fault (ID 9) and the Miaoli frontal structure (ID 10), the Meishan fault (ID 20) and the Chiayi frontal structure (ID 21), and the Chiayi frontal structure (ID 21) and the Tainan frontal structure (ID 41). Their findings were consistent with our results based on the Coulomb stress triggering.

Additionally, Shyu et al. (2020) suggested some other structure pairs for multiple-structure ruptures, such as the Shihtan fault (ID 13) and Tuntzuchiao fault (ID 15), the Houchiali fault (ID 25) and the Tainan frontal structure (ID 41), and the Chaochou fault (ID 29) and the Hengchun fault (ID 30). These pairs, however, do not fit our hypothesis. Take the Shihtan and Tuntzuchiao faults, for example. The rupture of the Tuntzuchiao fault could result in a Coulomb stress increase of more than 0.1 bar in 79% of the sub-faults of the Shihtan fault, whereas only 2% of the sub-fault in the Tuntzuchiao fault would be
triggered when the Shihtan fault dislocates (Table S1). Note that the 1935 Hsinchu-Taichung earthquake is attributed to a coseismic rupture on the two faults. Previous studies (Yan, 2016; Su, 2019) indicated that this earthquake did not initiate on either the Shihtan or the Tuntzuchiao fault, but on a blind fault linking the two. The database we accessed (Shyu et al., 2020) did not include this blind structure. Our analysis could be further improved through better understanding seismogenic structures. In addition, we discussed the interaction between structures through a kinematic model; it is desired to further
incorporate a dynamic model (e.g., Jiao et al., 2020) to constrain the behaviors of multiple-structure ruptures.

In 1906, an earthquake with magnitude 7.1 occurred due to the rupture of the Meishan fault (ID 20). Considering its fault geometry, the characteristic magnitude of this fault is only 6.6; therefore, this event with a larger magnitude could be associated with a multiple-structure rupture. Since liquefaction took place to the west of the Meishan fault during the coseismic period, the Chiayi frontal structure could also rupture simultaneously. The focal mechanism of the Chiayi frontal
structure is reverse faulting, while the Meishan fault is right-lateral strike-slip faulting, and the Meishan fault is northeast of the Chiayi frontal structure. Based on their rupture mechanisms and relative locations, our simplified model (Fig. 1b) suggested that an earthquake initiated on the Chiayi frontal structure could trigger a rupture on the Meishan fault, inferring their possible Coulomb stress interaction. We further considered detailed parameters of these two structures (i.e., structure geometry, characteristic slip) to quantify the possibility of a multiple-structure rupture. When the Meishan fault is dislocated,
the Coulomb stress on 64% of the Chiayi frontal structure plane may rise by more than 0.1 bar, and when the Chiayi frontal structure is dislocated, 72% of the Meishan fault could be closer to failure (Table S1). In addition, the distance between the two faults is 1.87 km (Table S2). Therefore, we concluded that these two structures could have mutually ruptured in a coseismic period and resulted in an event with magnitude 7.1 in 1906.

## 4.2 Uncertainty of the Coulomb stress model and recurrence interval

In this study, we identified potential rupture pairs by considering thresholds of stress change and structure distance. We implemented four threshold sets of Coulomb stress change (i.e., +0.01, +0.05, +0.1, and +0.2 bars) and two for distance





between structures (i.e., 2.5 and 5.0 km) to identify plausible pairs for multiple-structure rupture (Table 3). Also, the uncertainty of the structure rake angle could result in deviation. Our standard procedure assumed a fixed rake angle of each structure according to its rupture type (Table 1), while in reality, its rupture orientation could alter slightly in small patches

of the structure plane. To evaluate the impact of rake angle orientation, we evaluated the Coulomb stress change on receiver structure with different rotated rake angles (i.e., ±10˚ and ±20˚). The results showed that the larger the rotated rake angles implemented for receiver structures, the fewer structure pairs were identified (Table 5). Note that 11 pairs are identified even when the rakes rotate for ±20˚, suggesting its robustness for coseismic multiple-structure rupture. Besides the uncertainty of structure pair identification, rupture parameter uncertainties of the multiple structures could be

evaluated. Considering the range of the slip area of structures (Table 1), magnitude intervals of multiple-structure ruptures can be estimated (Table 2). That is, assuming the maximum slip areas of the two structures obtains the largest magnitude (based on equations 4-6). By further implementing structure slip rates, recurrence intervals can be quantified: the minimum slip area and maximum slip rate obtains the shortest recurrence interval (based on equations 4-17).

Based on our analyses, deviations of multiple-structure rupture pairs could be indicated, and uncertainties of corresponding

parameters were quantified, providing a better understanding of multiple-structure rupture behaviors, beneficial to subsequent research, such as the probabilistic seismic hazard assessment (PSHA), mentioned below.

**4.3 Application of multiple-structure rupture to probabilistic seismic hazard analysis**

Conducting a PSHA requires understanding the recurrence interval and potential magnitude of each seismogenic source, and implementing a hazard model with multiple-structure rupture could improve the assessment. Take the PSHA proposed by the

TEM in 2020 (TEM PSHA2020, Chan et al., 2020) as an example, considering the cases of multiple-structure ruptures, the hazard levels in the regions close to the Chaochou fault (ID 29) and the Tainan frontal structure (ID 41) increased significantly for a long return period (recurrence interval of 2,475 years, see Fig. 3 of Chan et al., 2020). Their study indicated that the seismic hazard level would be misestimated if the probability of multiple-structure rupture is not implemented.

Seismic hazard analysis plays an essential role for constructing infrastructures, such as nuclear power plants, that requires assuming a long return period. Thus, a seismogenic source with a long recurrence interval could be crucial for the analysis, raising the importance of multiple-fault rupture with a larger magnitude (larger than the characteristic earthquake of each individual structure).

The possibility of multiple-structure rupture used to be determined based on geological and geomorphological evidence with

subjective judgments. Our study implemented a Coulomb stress change combined with statistical approaches to indicate multiple-structure rupture pairs, which is transparent and reproducible.



In addition, our approach indicated various rupture pairs and quantified uncertainties. These outcomes could be incorporated into a PSHA through a logic tree. For example, larger weightings (possibilities) could be assumed for the pairs that fulfill more thresholds in the distance and Coulomb stress change (Table 3) and cases of rotated rake angles (Table 5). That

includes, for instance, the Shuanglianpo fault (ID 2) and the Hukou fault (ID 4); the Hukou fault (ID 4) and the Fengshan river strike-slip structure (ID 5); the Hsinchu fault (ID 6) and the Hsinchu frontal structure (ID 8); the Miaoli frontal structure (ID 10) and Tuntzuchiao fault (ID 15); the Muchiliao – Liuchia fault (ID 22) and the Chungchou structure (ID 23); and the Chishan fault (ID 26) and the Fengshan structure (ID 45).

### 4.4 Multiple structure rupture (with more than three structures)

The 2016 $M_w$7.8 Kaikōura, New Zealand, earthquake is an event resulting from ruptures on multiple structures. Shi et al. (2017) indicated that this earthquake included ruptures along four major faults and up to 12 minor faults. Based on this case, we are aware that multiple-structure rupture is not limited to the combination of two seismogenic structures.

Based on the multiple-structure rupture database proposed in this study (Table 2), several structures are associated with several possible rupture pairs. For instance, the Shuanglianpo fault (ID 2) may cause coseismic rupture with the Yangmei

structure (ID 3) and the Hukou fault (ID 4), and the Hukou fault (ID 4) may link with the Fengshan River strike-slip structure (ID 5) and the Hsinchu fault (ID 6). Since our approach is based on a static Coulomb stress change, it is difficult to evaluate the temporal evolution of rupture probability. The possibility of a multiple-structure rupture in a coseismic period might be overestimated. One potential solution is to implement a dynamic model (e.g., a discrete element model; Cundall and Strack, 1979) that simulates temporal distribution of displacement and stress fields and could be helpful in identifying

plausible structures that perhaps rupture within a coseismic period.

### 5 Acknowledgements

This study was supported by the Ministry of Science and Technology in Taiwan under the grants MOST 109-2116-M-008 - 029 -MY3 and MOST 110-2124-M-002 -008. This work is financially supported by the Earthquake-Disaster & Risk Evaluation and Management Center (E-DREaM) from the Featured Areas Research Center Program within the framework of

the Higher Education Sprout Project by the Ministry of Education in Taiwan.

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

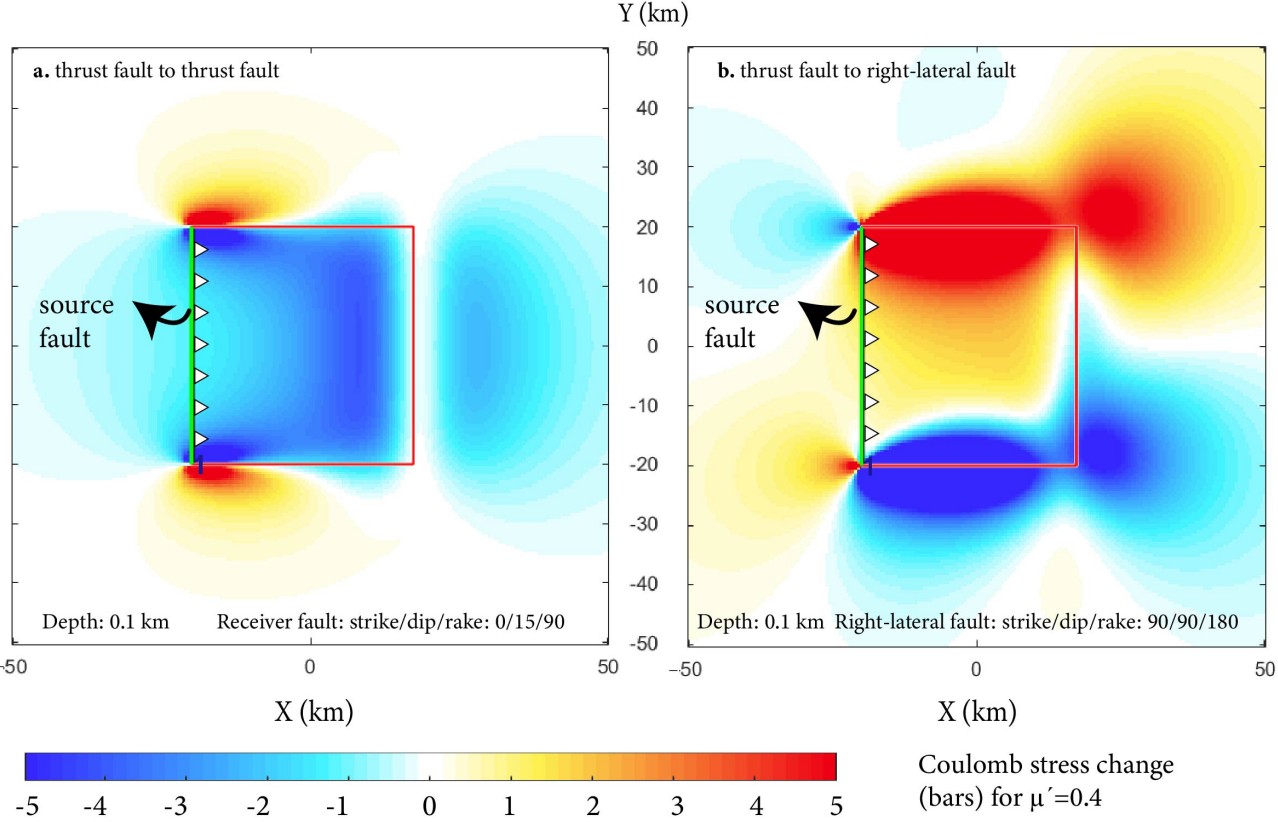

**Figure 1: (a) Distribution of the Coulomb stress change in a reverse mechanism environment, and (b) distribution of the Coulomb stress change by a reverse faulting event on a right-lateral mechanism environment.**




**Figure 2: Distribution of the 45 seismogenic structures in Taiwan. Corresponding structure parameters are listed in Table 1.**






## Table 1

| ID | Seismogenic structure name | Type | Rake | Depth 1 (km) | Dip (°) between depth 0 1 | Depth 2 (km) | Dip2 (°) between depth 1 2 | Depth 3 (km) | Dip3 (°) between depth 2 3 | Slip area (km²) | | | $M_w$* | Characteristic slip (m) | Slip rate (mm/year) | | |
|---|---|---|---|---|---|---|---|---|---|---|---|---|---|---|---|---|---|
| | | | | | | | | | | Minimum | Mean | Maximum | | | Minimum | Mean | Maximum |
| 1 | Shanchiao fault | N | -90 | 7.0 | 60 | 10.0 | 45 | 14 | 30 | 713.66 | 1053.50 | 1731.59 | 7.01 | 1.29 | 1.10 | 1.66 | 2.94 |
| 2 | Shuanglienpo structure | R | 90 | 3.0 | 45 | 5.0 | 15 | - | - | 79.20 | 131.67 | 237.93 | 6.24 | 0.72 | 0.06 | 0.13 | 0.52 |
| 3 | Yangmei structure | R | 90 | 3.0 | 60 | | | | | 47.07 | 76.47 | 115.36 | 6.03 | 0.60 | 0.09 | 0.18 | 0.72 |
| 4 | Hukou fault | R | 90 | 10.0 | 30 | - | - | - | - | 318.72 | 512.00 | 898.30 | 6.77 | 1.16 | 0.19 | 0.46 | 2.34 |
| 5 | Fengshan river strike slip structure | LL | 0 | 13.8 | 85 | | | | | 362.61 | 425.34 | 492.35 | 6.66 | 0.95 | 1.80 | 3.18 | 10.15 |
| 6 | Hsinchu fault | R | 90 | 10.0 | 45 | - | - | - | - | 141.67 | 205.03 | 303.34 | 6.41 | 0.83 | 0.28 | 0.66 | 2.91 |
| 7 | Hsincheng fault | R | 90 | 12.9 | 30 | - | - | - | - | 481.65 | 732.74 | 1238.33 | 6.91 | 1.31 | 0.47 | 1.12 | 5.35 |
| 8 | Hsinchu frontal structure | R | 90 | 10.0 | 30 | - | - | - | - | 150.65 | 242.00 | 424.59 | 6.48 | 0.90 | 0.62 | 1.44 | 6.81 |
| 9 | Touhuanping structure | RL | 180 | 12.0 | 85 | | | | | 258.00 | 310.89 | 366.88 | 6.52 | 0.80 | 0.12 | 0.13 | 0.14 |
| 10 | Miaoli frontal structure | R | 90 | 10.0 | 30 | - | - | - | - | 384.71 | 618.00 | 1084.28 | 6.84 | 1.22 | 0.78 | 1.84 | 8.77 |
| 11 | Tunglo structure | R | 90 | 3.5 | 30 | - | - | - | - | 61.07 | 109.90 | 206.61 | 6.17 | 0.68 | 0.19 | 0.50 | 2.63 |
| 12 | East Miaoli structure | R | 90 | 4.0 | 30 | - | - | - | - | 67.25 | 115.20 | 210.53 | 6.19 | 0.69 | 0.36 | 0.84 | 3.89 |
| 13 | Shihtan fault | R | 90 | 10.8 | 75 | - | - | - | - | 274.46 | 343.23 | 418.13 | 6.61 | 0.99 | 0.61 | 1.38 | 5.32 |
| 14 | Sanyi fault | R | 90 | 9.0 | 15 | - | - | - | - | 610.01 | 1036.15 | 1887.83 | 7.04 | 1.45 | 0.29 | 0.85 | 4.61 |
| 15 | Tuntzuchiao fault | RL | 180 | 14.8 | 85 | | | | | 345.33 | 400.95 | 460.35 | 6.64 | 0.94 | 0.27 | 0.50 | 1.7 |
| 16 | Changhua fault | R | 90 | 3.0 | 45 | 5.0 | 30 | 12 | 10 | 2036.09 | 3990.81 | 9799.06 | 7.57 | 2.35 | 0.95 | 1.87 | 6.97 |
| 17 | Chelungpu fault | R | 90 | 12.0 | 15 | - | - | - | - | 2687.16 | 4260.48 | 7408.98 | 7.60 | 2.45 | 6.94 | 6.94 | 6.94 |
| 18 | Tamaopu - Shuangtung fault | R | 90 | 6.0 | 30 | - | - | - | - | 538.38 | 830.40 | 1416.52 | 6.96 | 1.38 | 0.47 | 1.06 | 4.88 |
| 19 | Chiuchiungkeng fault | R | 90 | 12.0 | 30 | - | - | - | - | 522.82 | 806.40 | 1375.25 | 6.95 | 1.37 | 1.87 | 4.66 | 23.39 |
| 20 | Meishan fault | RL | 180 | 14.7 | 85 | | | | | 319.79 | 371.70 | 427.14 | 6.60 | 0.89 | 2.50 | 2.51 | 2.54 |
| 21 | Chiayi frontal structure | R | 90 | 12.0 | 15 | - | - | - | - | 997.08 | 1580.88 | 2749.14 | 7.21 | 1.71 | 1.40 | 3.36 | 16.12 |
| 22 | Muchiliao - Liuchia fault | R | 90 | 12.0 | 30 | - | - | - | - | 410.78 | 633.60 | 1080.55 | 6.85 | 1.23 | 4.40 | 5.75 | 7.1 |
| 23 | Chungchou structure | R | 90 | 12.0 | 30 | - | - | - | - | 454.35 | 700.80 | 1195.16 | 6.89 | 1.28 | 9.02 | 12.20 | 18.71 |
| 24 | Hsinhua fault | RL | 180 | 15.0 | 85 | - | - | - | - | 192.40 | 222.89 | 255.45 | 6.38 | 0.69 | 0.80 | 2.65 | 4.5 |
| 25 | Houchiali fault | R | 90 | 5.0 | 45 | | | | | 59.54 | 86.25 | 127.61 | 6.07 | 0.61 | 6.10 | 7.07 | 8.72 |
| 26 | Chishan fault | LL/R | 45 | 10.8 | 75 | | | | | 357.60 | 447.20 | 544.80 | 6.68 | 0.97 | 0.72 | 1.10 | 1.5 |
| 27 | Hsiaokangshan fault | R | 90 | 7.0 | 30 | | | | | 103.56 | 155.40 | 259.63 | 6.30 | 0.75 | 0.81 | 1.78 | 8.04 |
| 28 | Kaoping River structure | LL/R | 45 | 12.3 | 75 | - | - | - | - | 348.36 | 424.51 | 507.35 | 6.66 | 0.95 | 0.17 | 0.32 | 1.05 |
| 29 | Chaochou fault | LL/R | 45 | 11.1 | 75 | | | | | 918.53 | 1141.95 | 1385.24 | 7.10 | 1.62 | 0.57 | 0.98 | 3.01 |
| 30 | Hengchun fault | LL/R | 45 | 15.0 | 75 | - | - | - | - | 553.08 | 650.71 | 757.97 | 6.85 | 1.20 | 5.74 | 6.15 | 6.62 |
| 31 | Hengchun offshore structure | R | 90 | 4.0 | 30 | | | | | 92.00 | 157.60 | 288.01 | 6.31 | 0.77 | 1.87 | 3.22 | 6.96 |
| 32 | Milun fault | LL/R | 45 | 10.0 | 75 | - | - | - | - | 264.71 | 337.41 | 416.30 | 6.56 | 0.85 | 9.92 | 10.15 | 10.47 |
| 33 | Longitudinal Valley fault | R/LL | 45 | 5.0 | 75 | 15.0 | 60 | 20 | 45 | 2805.45 | 3509.02 | 4675.75 | 7.52 | 2.25 | 5.60 | 11.35 | 17.1 |
| 34 | Central Range structure | R | 90 | 20.0 | 45 | - | - | - | - | 1893.81 | 2437.74 | 3306.63 | 7.38 | 2.00 | 4.76 | 7.28 | 11.16 |
| 35 | Luyeh fault | R | 90 | 2.0 | 45 | 4.0 | 30 | | | 90.36 | 133.87 | 223.24 | 6.24 | 0.71 | 3.55 | 5.28 | 8.02 |
| 36 | Taimali coastline structure | R/LL | 45 | 10.6 | 75 | - | - | - | - | 373.24 | 469.99 | 574.48 | 6.73 | 1.10 | 5.74 | 7.32 | 9.03 |
| 37 | Northern Ilan structure | N | -90 | 9.4 | 60 | | | | | 590.96 | 814.16 | 1115.26 | 6.90 | 1.14 | 0.96 | 3.29 | 6.27 |
| 38 | Southern Ilan structure | N | -90 | 11.3 | 60 | | | | | 215.50 | 284.48 | 378.87 | 6.43 | 0.64 | 4.47 | 5.48 | 6.92 |
| 39 | Chushiang structure | R/RL | 135 | 3.0 | 55 | | | | | 43.76 | 72.47 | 112.07 | 6.00 | 0.57 | 2.03 | 5.01 | 9 |
| 40 | Gukeng structure | LL | 0 | 12.0 | 85 | | | | | 92.00 | 110.86 | 130.82 | 6.07 | 0.48 | 0.56 | 0.94 | 2.56 |
| 41 | Tainan frontal structure | R | 90 | 3.0 | 30 | 12.0 | 15 | | | 1076.49 | 1722.64 | 2963.96 | 7.24 | 1.74 | 0.45 | 0.92 | 3.5 |
| 42 | Longchuan structure | R | 90 | 12.0 | 60 | - | - | - | - | 245.78 | 320.17 | 422.27 | 6.58 | 0.96 | 0.85 | 1.73 | 6.53 |
| 43 | Youchang sturcture | R/RL | 135 | 12.0 | 75 | | | | | 171.81 | 216.17 | 256.47 | 6.41 | 0.83 | 0.92 | 1.64 | 5.46 |
| 44 | Fengshan hills frontal structure | R | 90 | 15.0 | 30 | | | | | 386.20 | 573.00 | 949.27 | 6.81 | 1.19 | 0.4 | 0.92 | 4.24 |
| 45 | Fengshan structure | LL/R | 30 | 15.0 | 85 | | | | | 218.40 | 253.01 | 289.97 | 6.50 | 1.19 | 10.00 | 10.00 | 10.00 |

*Obtained through a scaling law by considering mean slip area

**Table 1: The structure parameters of the 45 seismogenic structures in Taiwan. The alignments of the structures are presented in Figure 2. LL: left-lateral strike-slip mechanism; N: normal mechanism; R: reverse mechanism; RL: right-lateral strike-slip mechanism.**






Table 2

| ID | Seismogenic structure name | Type | with minimum area | | Recurrence interval (year) | | | with mean area | | Recurrence interval (year) | | | with maximum area | | Recurrence interval (year) | | |
|---|---|---|---|---|---|---|---|---|---|---|---|---|---|---|---|---|---|
| | | | Area (km) | $M_w$ | for min slip rate | for mean slip rate | for max slip rate | Area (km) | $M_w$ | for min slip rate | for mean slip rate | for max slip rate | Area (km) | $M_w$ | for min slip rate | for mean slip rate | for max slip rate |
| 2, 3 | Shuanglienpo structure, Yangmei structure | R, R | 126.27 | 6.22 | 20647 | 10029 | 2489 | 208.14 | 6.42 | 27419 | 13281 | 3346 | 353.29 | 6.62 | 37000 | 18500 | 4604 |
| 2, 4 | Shuanglienpo structure, Hukou fault | R, R | 397.92 | 6.67 | 23444 | 9953 | 2010 | 643.67 | 6.86 | 29929 | 12324 | 2499 | 1136.23 | 7.08 | 39026 | 16191 | 3287 |
| 4, 5 | Hukou fault, Fengshan river strike-slip structure | R, LL | 681.33 | 6.88 | 2141 | 1192 | 360 | 937.34 | 7.00 | 2794 | 1550 | 464 | 1390.65 | 7.16 | 4098 | 2254 | 668 |
| 4, 6 | Hukou fault, Hsinchu fault | R, R | 460.39 | 6.73 | 18377 | 7574 | 1586 | 717.03 | 6.90 | 21949 | 9250 | 1930 | 1201.64 | 7.10 | 28556 | 11953 | 2495 |
| 6, 8 | Hsinchu fault, Hsinchu frontal structure | R, R | 292.32 | 6.55 | 4000 | 1721 | 368 | 447.03 | 6.72 | 5096 | 2184 | 467 | 727.93 | 6.91 | 6809 | 2929 | 626 |
| 6, 9 | Hsinchu fault, Touhuanping structure | R, RL | 399.67 | 6.63 | 16926 | 9723 | 2874 | 515.92 | 6.75 | 20226 | 11527 | 3268 | 670.22 | 6.86 | 24140 | 13120 | 3636 |
| 9, 10 | Touhuanping structure, Miaoli frontal structure | RL, R | 642.71 | 6.86 | 6423 | 2881 | 630 | 928.89 | 7.00 | 7204 | 3209 | 695 | 1451.16 | 7.18 | 8858 | 3914 | 842 |
| 10, 15 | Miaoli frontal structure, Tuntzuchiao fault | R, R | 730.04 | 6.91 | 5510 | 2513 | 572 | 1018.95 | 7.04 | 6371 | 2870 | 643 | 1544.63 | 7.20 | 7811 | 3473 | 769 |
| 11, 14 | Tunglo structure, Sanyi fault | R, R | 671.08 | 6.87 | 11664 | 4000 | 741 | 1146.05 | 7.08 | 15090 | 5276 | 975 | 2094.44 | 7.32 | 21747 | 7478 | 1387 |
| 13, 14 | Shihtan fault, Sanyi fault | R, R | 884.47 | 6.98 | 6920 | 2735 | 598 | 1379.38 | 7.16 | 9667 | 3757 | 806 | 2305.96 | 7.36 | 14093 | 5391 | 1135 |
| 19, 22 | Chiuchiungkeng fault, Muchiliao - Liuchia fault | R, R | 933.60 | 7.00 | 998 | 539 | 151 | 1440.00 | 7.17 | 1270 | 691 | 196 | 2455.80 | 7.38 | 1755 | 965 | 278 |
| 20, 21 | Meishan fault, Chiayi frontal structure | RL, R | 1316.87 | 7.14 | 2104 | 1251 | 345 | 1952.58 | 7.29 | 2722 | 1553 | 409 | 3176.28 | 7.48 | 3871 | 2097 | 527 |
| 21, 41 | Chiayi frontal structure, Tainan frontal structure | R, R | 2073.57 | 7.32 | 4475 | 1966 | 438 | 3303.52 | 7.50 | 5726 | 2512 | 558 | 5713.10 | 7.71 | 7776 | 3402 | 755 |
| 22, 23 | Muchiliao - Liuchia fault, Chungchou structure | R, R | 865.13 | 6.97 | 364 | 271 | 184 | 1334.40 | 7.14 | 471 | 351 | 239 | 2275.71 | 7.35 | 663 | 494 | 337 |
| 24, 25 | Hsinhua fault, Houchiali fault | RL, R | 251.94 | 6.43 | 559 | 326 | 222 | 309.14 | 6.52 | 609 | 367 | 254 | 383.06 | 6.61 | 661 | 413 | 288 |
| 26, 45 | Chishan fault, Fengshan structure | LL/R, LL/R | 576.00 | 6.80 | 615 | 573 | 534 | 742.38 | 6.91 | 706 | 661 | 619 | 834.77 | 6.96 | 825 | 766 | 713 |
| 43, 45 | Youchang sturcture, Fengshan structure | R/RL, LL/R | 390.21 | 6.62 | 405 | 374 | 265 | 501.35 | 6.73 | 465 | 432 | 314 | 546.44 | 6.77 | 530 | 487 | 341 |

**Table 2: Potential pairs of multiple-structure ruptures, their parameters, and recurrence intervals of earthquakes. LL: left-lateral strike-slip mechanism; R: reverse mechanism; RL: right-lateral strike-slip mechanism.**






Table 3

| ID | Seismogenic structure name | 5.0 km | | | | 2.5 km | | | | Max. distance between a pair |
|---|---|---|---|---|---|---|---|---|---|---|
| | | 0.01 bar | 0.05 bar | 0.1 bar | 0.2 bar | 0.01 bar | 0.05 bar | 0.1 bar | 0.2 bar | ΔCFS triggering threshold |
| 2, 3 | Shuanglienpo structure, Yangmei structure | ✓ | ✓ | ✓ | ✓ | ✓ | ✓ | ✓ | | |
| 2, 4 | Shuanglienpo structure, Hukou fault | ✓ | ✓ | ✓ | ✓ | ✓ | ✓ | ✓ | ✓ | |
| 4, 5 | Hukou fault, Fengshan river strike-slip structure | ✓ | ✓ | ✓ | ✓ | ✓ | ✓ | ✓ | ✓ | |
| 4, 6 | Hukou fault, Hsinchu fault | ✓ | ✓ | ✓ | | ✓ | ✓ | ✓ | | |
| 4, 8 | Hukou fault, Hsinchu frontal structure | ✓ | ✓ | | | ✓ | | | | |
| 6, 8 | Hsinchu fault, Hsinchu frontal structure | ✓ | ✓ | ✓ | ✓ | ✓ | ✓ | ✓ | ✓ | |
| 6, 9 | Hsinchu fault, Touhuanping structure | ✓ | ✓ | ✓ | ✓ | ✓ | | | | |
| 9, 10 | Touhuanping structure, Miaoli frontal structure | ✓ | ✓ | ✓ | | ✓ | ✓ | ✓ | | |
| 10, 15 | Miaoli frontal structure, Tuntzuchiao fault | ✓ | ✓ | ✓ | ✓ | ✓ | ✓ | ✓ | ✓ | |
| 10, 16 | Miaoli frontal structure, Changhua fault | ✓ | ✓ | | | ✓ | ✓ | | | |
| 11, 14 | Tunglo structure, Sanyi fault | ✓ | ✓ | ✓ | | ✓ | ✓ | ✓ | | |
| 11, 16 | Tunglo structure, Changhua fault | ✓ | | | | ✓ | | | | |
| 13, 14 | Shihtan fault, Sanyi fault | ✓ | ✓ | ✓ | | ✓ | ✓ | ✓ | | |
| 13, 16 | Shihtan fault, Changhua fault | ✓ | | | | ✓ | | | | |
| 14, 17 | Sanyi fault, Chelungpu fault | ✓ | ✓ | | | ✓ | ✓ | | | |
| 15, 16 | Tuntzuchiao fault, Changhua fault | ✓ | | | | ✓ | | | | |
| 16, 19 | Changhua fault, Chiuchiungkeng fault | ✓ | ✓ | | | ✓ | ✓ | | | |
| 16, 20 | Changhua fault, Meishan fault | ✓ | | | | ✓ | | | | |
| 16, 40 | Changhua fault, Gukeng structure | ✓ | | | | | | | | |
| 17, 19 | Chelungpu fault, Chiuchiungkeng fault | ✓ | | | | ✓ | | | | |
| 17, 20 | Chelungpu fault, Meishan fault | ✓ | | | | ✓ | | | | |
| 17, 40 | Chelungpu fault, Gukeng structure | ✓ | | | | | | | | |
| 19, 22 | Chiuchiungkeng fault, Muchiliao - Liuchia fault | ✓ | ✓ | ✓ | ✓ | ✓ | | | | |
| 20, 21 | Meishan fault, Chiayi frontal structure | ✓ | ✓ | ✓ | | ✓ | ✓ | ✓ | | |
| 21, 41 | Chiayi frontal structure, Tainan frontal structure | ✓ | ✓ | ✓ | | ✓ | | | | |
| 22, 23 | Muchiliao - Liuchia fault, Chungchou structure | ✓ | ✓ | ✓ | ✓ | ✓ | ✓ | ✓ | ✓ | |
| 23, 27 | Chungchou structure, Hsiaokangshan fault | ✓ | ✓ | | | ✓ | ✓ | | | |
| 24, 25 | Hsinhua fault, Houchiali fault | ✓ | ✓ | ✓ | | ✓ | ✓ | ✓ | | |
| 24, 41 | Hsinhua fault, Tainan frontal structure | ✓ | | | | ✓ | | | | |
| 26, 45 | Chishan fault, Fengshan structure | ✓ | ✓ | ✓ | ✓ | ✓ | ✓ | ✓ | ✓ | |
| 27, 42 | Hsiaokangshan fault, Longchuan structure | ✓ | ✓ | | | ✓ | ✓ | | | |
| 30, 31 | Hengchun fault, Hengchun offshore structure | ✓ | | | | | | | | |
| 32, 33 | Milun fault, Longitudinal Valley fault | ✓ | | | | ✓ | | | | |
| 43, 45 | Youchang sturcture, Fengshan structure | ✓ | ✓ | ✓ | ✓ | ✓ | | | | |
| | Total pairs of each criteria | 34 | 23 | 17 | 10 | 31 | 18 | 13 | 6 | |

**Table 3: Multiple-structure rupture pairs considering different thresholds in structure distance and Coulomb stress change.**






Table 4

| ID | Seismogenic structure name | Type | Original slip rate (mm/year) | Remained slip rate (mm/year) | Original recurrence interval (year) | Updated recurrence interval (year) |
|---|---|---|---|---|---|---|
| 2 | Shuanglienpo structure | R | 0.13 | 0.033 | 5540 | 21818 |
| 3 | Yangmei structure | R | 0.18 | 0.074 | 3330 | 8106 |
| 4 | Hukou fault | R | 0.46 | 0.104 | 2520 | 11154 |
| 5 | Fengshan river strike-slip structure | SS | 3.18 | 1.337 | 300 | 710 |
| 6 | Hsinchu fault | R | 0.66 | 0.125 | 1260 | 6640 |
| 8 | Hsinchu frontal structure | R | 1.44 | 0.642 | 1170 | 1401 |
| 9 | Touhuanping structure | SS | 0.13 | 0.034 | 6150 | 23529 |
| 10 | Miaoli frontal structure | R | 1.84 | 0.547 | 660 | 2230 |
| 11 | Tunglo structure | R | 0.50 | 0.151 | 1360 | 4509 |
| 13 | Shihtan fault | R | 1.38 | 0.519 | 720 | 1908 |
| 14 | Sanyi fault | R | 0.85 | 0.269 | 1710 | 5390 |
| 15 | Tuntzuchiao fault | SS | 0.50 | 0.204 | 1880 | 4601 |
| 19 | Chiuchiungkeng fault | R | 4.66 | 2.093 | 290 | 503 |
| 20 | Meishan fault | SS | 2.51 | 0.871 | 350 | 1059 |
| 21 | Chiayi frontal structure | R | 3.36 | 0.992 | 510 | 1724 |
| 22 | Muchiliao - Liuchia fault | R | 5.75 | 1.573 | 210 | 782 |
| 23 | Chungchou structure | R | 12.2 | 5.393 | 100 | 237 |
| 24 | Hsinhua fault | SS | 2.65 | 1.238 | 260 | 557 |
| 25 | Houchiali fault | R | 7.07 | 2.806 | 90 | 217 |
| 26 | Chishan fault | SS/R | 1.10 | 0.492 | 880 | 1971 |
| 41 | Tainan frontal structure | R | 0.92 | 0.405 | 1890 | 4294 |
| 43 | Youchang sturcture | R/SS | 1.64 | 0.699 | 510 | 1188 |
| 45 | Fengshan structure | SS/R | 10.00 | 2.604 | 75 | 288 |

**Table 4: Original and revised recurrence intervals of the seismogenic structures that involve the cases of multiple-structure rupture. LL: left-lateral strike-slip mechanism; N: normal mechanism; R: reverse mechanism; RL: right-lateral strike-slip mechanism.**



## Table 5

| Rake angle rotation | +10° | -10° | +20° | -20° |
|---|---|---|---|---|
| | 2 3 | 2 3 | 2 3 | 2 3 |
| | 2 4 | 2 4 | 2 4 | 2 4 |
| | 4 5 | 4 5 | 4 5 | 4 5 |
| | 4 6 | 4 6 | 4 6 | 4 6 |
| | 6 8 | 6 8 | 6 8 | 6 8 |
| | 6 9 | 6 9 | 6 9 | 6 9 |
| | 9 10 | 9 10 | 9 10 | 9 10 |
| Paired structures at each specific rake condition | 10 15 | 10 15 | 10 15 | 10 15 |
| | 11 14 | 11 14 | 11 14 | 11 14 |
| | 13 14 | 13 14 | 13 14 | 13 14 |
| | 19 22 | 19 22 | 19 22 | 19 22 |
| | 20 21 | 20 21 | 20 21 | 20 21 |
| | 21 41 | 21 41 | 21 41 | 21 41 |
| | 22 23 | 22 23 | 22 23 | 22 23 |
| | 24 25 | 24 25 | 24 25 | 24 25 |
| | 26 45 | 26 45 | 26 45 | 26 45 |
| | 43 45 | 43 45 | 43 45 | 43 45 |
| Number of pair | 16 | 15 | 13 | 11 |

Number of pairs without rake angle rotation: 17

| | | |
|---|---|---|
| 2 3 | Paired structures at the condition | |
| 2 3 | Not paired structures at the condition | |

**Table 5: Potential paired structures considering various rake angle rotations. In these cases, the stress threshold of ΔCFS ⩾ 0.1**
**and distance threshold of 5 km were considered to identify potential rupture pairs. The total number of paired structures without rake rotation is 17 (Table 2).**