# Peer review of "Quantifying the probability and uncertainty of multiple-structure rupture and recurrence intervals in Taiwan"

_Natural Hazards and Earth System Sciences, 2022_

## Referee Comment (RC1)

This study by Chang et al uses Coulomb stress modelling and the power-law distributions of earthquake magnitudes implied by the Gutenberg-Richter (G-R) relationship to determine if, and how frequently, closely spaced faults in the Taiwan Earthquake Model (TEM) will rupture together in multi-fault earthquakes. The need for this kind of analysis has been raised by the recent occurrence of multifault earthquakes (e.g., 2010 El Mayor-Cucapah and 2016 Kaikōura earthquakes). Furthermore, the ground motions derived from probabilistic seismic hazard analysis (PSHA) are very sensitive to the assumed on-fault magnitude-frequency distribution. The model outlined here provides a logical way to account for the discrete possibilities of faults (referred to here as 'structures') rupturing either along their entire length, or in multiple-structure ruptures. This work therefore clearly fits within the scope of NHESS.

I do, however, have some suggestions for how this study can be improved through a better description of how slip rates are partitioned between different rupture scenarios. I also recommend this study should be properly paced in the context of many other methods that have been developed to account for multi-fault ruptures in PSHA. I hope the authors find this review useful.

Jack Williams

Major Comments

1.) Description of model

The key innovation of this study is described in Section 3.1 where it is outlined how the area and slip rate (i.e., a moment rate) of two different seismogenic structures can be combined with a G-R relationship to determine the recurrence interval of an earthquake that ruptures both structures. As far as I can tell, there is nothing inherently wrong with the approach itself, however, I have several recommendations for how the presentation of this model could be improved.

Immediately after equations 8 and 9 (Line 129), the meaning $\dot{D}_{L1}$ (original $L_1$ slip rate measurement) is given, but it is $\dot{D}_{L1}'$ (slip rate for $L_1$ single structure events) that is used in these equations. I would also present the equations for $C_1$ (partitioning coefficient between $\dot{D}_{L1+L2}^{L1}$ and $\dot{D}_{L1}'$, currently eqs. 10 and 11) and $\dot{D}_{L1}'$ (currently eqs. 12 and 13) before the equation for $\dot{D}_{L1+L2}^{L1}$ (slip rate of $L_1$ in $L_1+L_2$ events) given that you need these parameters to calculate $\dot{D}_{L1+L2}^{L1}$.

I appreciate that the authors use the Hsinhua and Houchiali faults to provide an example of how their workflow is applied. However, showing the application of each equation to each structure in the text can get repetitive. I would suggest using a table to illustrate these equations, with a column for each structure. Another table could also be used for description of the model where >2 structures are considered (i.e., the example of the Chiayi, Meishan, and Tainan structures in Section 3.2).

I would also recommend adding a table (maybe as a supplementary file) illustrating that when the slip rate is partitioned between the different rupture cases, the total seismic moment rate does not change. I provide an example of this below

| | Analysis using original slip rate estimates | | | Analysis using partitioned slip rate estimates | | | |
| --- | --- | --- | --- | --- | --- | --- | --- |
| | Hsinhua | Houciali | | Hsinhua &Houciali | Hsinhua | Houciali | |
| Area (m^2) | 229000000 | 86000000 | | 309140000 | 229000000 | 86000000 | |
| Rigidity (Nm) | 30000000000 | 30000000000 | | 30000000000 | 30000000000 | 30000000000 | |
| Slip Rate (m/yr) | 0.00265 | 0.00707 | **Total Moment Rate** | 0.00220 | 0.00124 | 0.002822 | **Total Moment Rate** |
| Seismic Moment Rate (Nm/yr) | 1.82E+16 | 1.82E+16 | **3.64E+16** | 2.04E+16 | 8.52E+15 | 7.28E+15 | **3.62E+16** |

This table shows that the moment rate ($\dot{M}_0$) of the two seismogenic structures using the original slip rate estimates is essentially the same as when the slip rate is partitioned between single structure and multi-structure events (I haven't checked but presumably the minor difference in total $\dot{M}_0$ can be accounted for in rounding errors). Hence, it gives the reader confidence that no $\dot{M}_0$ is being lost or gained when this model is applied.

When using the examples from the TEM, the values are provided to a high, and probably unjustified level of specificity (e.g. slip rates to 0.01 mm/yr, source areas to 0.01 km$^2$, recurrence intervals to 1 year). I suggest rounding these values to a level appropriate with the uncertainty of this analysis.

Finally, the Wells and Coppersmith (1994) scaling relationships are increasingly out of date given that we now have nearly 30 more years' worth of observed earthquakes to refine these relationships. I would recommend that either a more up to date set of scaling relationships is used (e.g., Leonard 2010, Thingbaijam et al 2017), or a sensitivity analysis is made to see if using the updated scaling relationships changes the model outcomes.

2.) Applicability of Model

I also have several comments about applicability of this model to observed occurrences of multi-structure earthquakes. However, I see these as points that can be addressed through additions to the discussion (Section 4) as opposed to changes to the model itself.

Multi-structure earthquakes are considered here only in terms of static Coulomb stress triggering between neighbouring faults. However, it is worth acknowledging in Section 4.2 that multi-structure earthquakes may also be generated by dynamic stress triggering from seismic waves (e.g., Brodsky and van der Elst 2014, Ulrich et al 2018). I think this may what is being discussed at Line 280 (?), though note the reference is to a manuscript (Jiao et al 2020) that was not accepted for publication.

A key assumption in this study is that the magnitude-frequency distribution (MFD) of events along a single multi-structure systems follow a G-R scaling. Although that is certainly possible, one could also argue that at the scale of a single multi-structure system, the MFD follows a characteristic shape (Youngs and Coppersmith 1984; Hecker et al 2013; Stirling and

Zungia 2017), or that the MFD is neither characteristic nor G-R (Geist and Parsons 2019; Page et al 2021). In either case, a deviation from a G-R scaling will affect the recurrence intervals calculated through this model.

This model should also be discussed in the context of other studies that have attempted to incorporate multi-structure ruptures in PSHA. For example, there are many studies that divide mapped multi-structure systems into smaller sub-fault scale segments, and then essentially allow ruptures to 'float' across theses smaller segments in such a way that they fit a regional MFD target (Field et al 2014; 2021; Chartier et al 2019; Geist and Parsons 2019). These studies are therefore distinct from the model described here, which is quite prescriptive about the number of configurations that structures in the TEM can rupture in (i.e., as single or multi-structure events only, and no events may be smaller than a single structure). It would benefit this study if the pros and cons of these different techniques could be discussed in Section 4.3.

Minor Comments

Lines 7-21: The abstract does not mention that this study is using faults incorporated into the Taiwan Earthquake Model to perform this analysis. Suggest revise, Line 11 could be revised to:

'……the probability of Coulomb stress triggering between seismogenic structures included in the Taiwan Earthquake Model.'

Lines 64-68: What value is used for the effective coefficient of friction ($\mu'$) in the Coulomb stress modelling?

Line 115: These scaling relationships between magnitude and rupture area are presumably from Wells and Coppersmith (1994)? If so, they should be cited as such (though also see major comment #1)

Line 144: Replace 'integrating' with 'combining,' to avoid any connotations that you are actually performing an integration in these equations.

Lines 220-221 (and 335): When referring to the Kaikōura earthquake, reference should be made to Hamling et al (2017). This is the original reference to this event and written by authors who made the primary observations of this multi-fault earthquake.

Line 258: I think there is a typo here for describing the numeric value if the Hukou and Hsinchu fault recurrence intervals as '4.4 and 5.3'?

Figures: Figure 1 presents only a generic case of Coulomb stress changes around a fault. I would recommend also including a figure to show an example of this stress modelling from faults in the TEM. Maybe using the example of faults that are described further in Section 3.1?

**References**

- Brodsky, E. E., & van der Elst, N. J. (2014). The uses of dynamic earthquake triggering. *Annual Review of Earth and Planetary Sciences*, *42*, 317-339.
- Chartier, T., Scotti, O., & Lyon-Caen, H. (2019). SHERIFS: Open-source code for computing earthquake rates in fault systems and constructing hazard models. *Seismological Research Letters*, *90*(4), 1678-1688.
- Field, E. H., Arrowsmith, R. J., Biasi, G. P., Bird, P., Dawson, T. E., Felzer, K. R., ... & Zeng, Y. (2014). Uniform California earthquake rupture forecast, version 3 (UCERF3)—The time-independent model. *Bulletin of the Seismological Society of America*, *104*(3), 1122-1180.
- Field, E. H., Milner, K. R., & Page, M. T. (2021). Generalizing the inversion-based PSHA source model for an interconnected fault system. *Bulletin of the Seismological Society of America*, *111*(1), 371-390.
- Geist, E. L., & Parsons, T. (2019). A combinatorial approach to determine earthquake magnitude distributions on a variable slip-rate fault. *Geophysical Journal International*, *219*(2), 734-752.
- Hamling, I. J., Hreinsdóttir, S., Clark, K., Elliott, J., Liang, C., Fielding, E., ... & Stirling, M. (2017). Complex multifault rupture during the 2016 M w 7.8 Kaikōura earthquake, New Zealand. *Science*, *356*(6334), eaam7194.
- Hecker, S., Abrahamson, N. A., & Wooddell, K. E. (2013). Variability of displacement at a point: Implications for earthquake-size distribution and rupture hazard on faults. *Bulletin of the Seismological Society of America*, *103*(2A), 651-674.
- Jiao, L., Chan, C. H., Scholtès, L., Hubert-Ferrari, A., Donzé, F. V., & Tapponnier, P. (2020). Discrete element modeling of a subduction zone with a seafloor irregularity and its impact on the seismic cycle. *Solid Earth Discussions*, 1-41.
- Leonard, M. (2010). Earthquake fault scaling: Self-consistent relating of rupture length, width, average displacement, and moment release. *Bulletin of the Seismological Society of America*, *100*(5A), 1971-1988.
- Page, M. T. (2021). More fault connectivity is needed in seismic hazard analysis. *Bulletin of the Seismological Society of America*, *111*(1), 391-397.
- Stirling, M. W., & Zuniga, F. R. (2017). Shape of the magnitude–frequency distribution for the Canterbury earthquake sequence from integration of seismological and geological data. *Bulletin of the Seismological Society of America*, *107*(1), 495-500.
- Thingbaijam, K. K. S., Martin Mai, P., & Goda, K. (2017). New empirical earthquake source-scaling laws. *Bulletin of the Seismological Society of America*, *107*(5), 2225-2246.
- Ulrich, T., Gabriel, A. A., Ampuero, J. P., & Xu, W. (2019). Dynamic viability of the 2016 Mw 7.8 Kaikōura earthquake cascade on weak crustal faults. *Nature communications*, *10*(1), 1-16.
- Wells, D. L., & Coppersmith, K. J. (1994). New empirical relationships among magnitude, rupture length, rupture width, rupture area, and surface displacement. *Bulletin of the seismological Society of America*, *84*(4), 974-1002.
- Youngs, R. R., & Coppersmith, K. J. (1985). Implications of fault slip rates and earthquake recurrence models to probabilistic seismic hazard estimates. *Bulletin of the Seismological society of America*, *75*(4), 939-964.

---

## Author Comment (AC1)

**Response to Reviewer #1 Jack Williams**

We greatly appreciate the reviewer's insightful comments and have revised our manuscript, nhess-2022-46, entitled, "Quantifying the probability and uncertainty of multiple-structure rupture and recurrence intervals in Taiwan," accordingly. Below, we have quoted the comments in italics and provided our detailed responses. All the changes are underlined in the revised manuscript.

*1.) Description of model*

*The key innovation of this study is described in Section 3.1 where it is outlined how the area and slip rate (i.e., a moment rate) of two different seismogenic structures can be combined with a G-R relationship to determine the recurrence interval of an earthquake that ruptures both structures. As far as I can tell, there is nothing inherently wrong with the approach itself, however, I have several recommendations for how the presentation of this model could be improved.*

We appreciate the reviewer's very helpful recommendations. We considered the reviewer's comments and responded in the following.

*Immediately after equations 8 and 9 (Line 129), the meaning $D_{L1}$ (original L1 slip rate measurement) is given, but it is $D_{L1}'$ (slip rate for L1 single structure events) that is used in these equations. I would also present the equations for C1 (partitioning coefficient between $D_{L1+L2L1}$ and $D_{L1}'$, currently eqs. 10 and 11) and $D_{L1}'$ (currently eqs. 12 and 13) before the equation for $D_{L1+L2L1}$ (slip rate of L1 in L1+L2 events) given that you need these parameters to calculate $D_{L1+L2L1}$.*

To clearly describe our algorithm for evaluating recurrence interval of multiple-structure ruptures, we first introduced the slip rate partitioned to individual structure ruptures (equations 8 and 9), followed by the obtained partitioned rates (equations 10 and 11). By combining them, the slip rate partitioned to the multiple-structure rupture from the original structures can be obtained (described in lines 123-140).

*I appreciate that the authors use the Hsinhua and Houchiali faults to provide an example of how their workflow is applied. However, showing the application of each equation to each structure in the text can get repetitive. I would suggest using a table to illustrate these equations, with a column for each structure. Another table could also be used for description of the model where >2 structures are considered (i.e., the example of the Chiayi, Meishan, and Tainan structures in Section 3.2).*

In the previous manuscript, this example is provided to demonstrate the procedure of the workflow. To simplify the description of the calculation, this example has been

removed.

*When using the examples from the TEM, the values are provided to a high, and probably unjustified level of specificity (e.g. slip rates to 0.01 mm/yr, source areas to 0.01 km2, recurrence intervals to 1 year). I suggest rounding these values to a level appropriate with the uncertainty of this analysis.*

We followed the reviewer's comment and revised Table 1 accordingly. Now the slip rate and slip area are rounded to one decimal place and the nearest whole number, respectively. Note that we keep recurrence intervals to 1 year, since some structures (e.g., the Milun fault) obtain short recurrence intervals (<100 years).

*Finally, the Wells and Coppersmith (1994) scaling relationships are increasingly out of date given that we now have nearly 30 more years' worth of observed earthquakes to refine these relationships. I would recommend that either a more up to date set of scaling relationships is used (e.g., Leonard 2010, Thingbaijam et al 2017), or a sensitivity analysis is made to see if using the updated scaling relationships changes the model outcomes.*

To validate the sensitivity of our procedure to scaling, we also implemented alternative relationships proposed by Yen and Ma (2011), who investigated the rupture parameters of the earthquakes mainly from the Taiwan orogenic belt. Based on this relation, recurrence intervals for each multiple-structure rupture pairs were evaluated (Table 5). Comparing these with those obtained by Wells and Coppersmith's relations, shorter recurrence intervals were obtained, especially for those with larger magnitude. These results can be attributed to a smaller average displacement obtained for a large event that led to a shorter recurrence interval for the multiple-structure rupture (based on equation 17). Note that although the scaling relations proposed by Wells and Coppersmith (1994) have been questioned by many modern models, especially for large megathrusts, Wang et al. (2016[b]) concluded similar maximal magnitude of each seismogenic structure estimated from the relations of Wells and Coppersmith (1994) and Yen and Ma (2011). We provided more detailed descriptions in lines 214-223, 292-298.

*Multi-structure earthquakes are considered here only in terms of static Coulomb stress triggering between neighbouring faults. However, it is worth acknowledging in Section 4.2 that multi-structure earthquakes may also be generated by dynamic stress triggering from seismic waves (e.g., Brodsky and van der Elst 2014, Ulrich et al 2018). I think this may what is being discussed at Line 280 (?), though note the reference is to a manuscript (Jiao et al 2020) that was not accepted for publication.*

We followed the reviewer's comment and indicated dynamic models could also constrain the behaviors of multiple-structure ruptures (lines 242-245). Note that the paper by Jiao et al. has been published in 2022.

*A key assumption in this study is that the magnitude-frequency distribution (MFD) of events along a single multi-structure systems follow a G-R scaling. Although that is certainly possible, one could also argue that at the scale of a single multi-structure system, the MFD follows a characteristic shape (Youngs and Coppersmith 1984; Hecker et al 2013; Stirling and Zungia 2017), or that the MFD is neither characteristic nor G-R (Geist and Parsons 2019; Page et al 2021). In either case, a deviation from a G-R scaling will affect the recurrence intervals calculated through this model.*

We are aware of the importance of the magnitude-frequency distribution (MFD) on a single-structure rupture, and the MFD could be in various forms, including the Gutenberg-Richter law and the characteristic earthquake model. In this study, we evaluated the rupture recurrence interval as the ratio of slip of a characteristic earthquake (with maximum magnitude of the structure) and slip rate based on the assumption proposed by the TEM seismogenic structure database and the TEM PSHA2020. Note that this factor could be replaced by other magnitude-frequency distributions since the recurrence interval of the multiple-structure rupture in our procedure is based on slip rate partitioned from individual structure ruptures (shown as equations 8-9, 14, 18, and 20). We provided more detailed descriptions in lines 101-104, 299-307.

*This model should also be discussed in the context of other studies that have attempted to incorporate multi-structure ruptures in PSHA. For example, there are many studies that divide mapped multi-structure systems into smaller sub-fault scale segments, and then essentially allow ruptures to 'float' across theses smaller segments in such a way that they fit a regional MFD target (Field et al 2014; 2021; Chartier et al 2019; Geist and Parsons 2019). These studies are therefore distinct from the model described here, which is quite prescriptive about the number of configurations that structures in the TEM can rupture in (i.e., as single or multi-structure events only, and no events may be smaller than a single structure). It would benefit this study if the pros and cons of these different techniques could be discussed in Section 4.3.*

Based on the assumption of the TEM PSHA2020, every rupture on a seismogenic structure results in a characteristic earthquake (with maximum magnitude of the structure), that is, small earthquakes (with magnitude smaller than the maximum magnitude of the structure) are attributed to shallow background sources. Following

this assumption, we did not consider ruptures on small segments of a structure.

*Minor Comments*
*Lines 7-21: The abstract does not mention that this study is using faults incorporated into the Taiwan Earthquake Model to perform this analysis. Suggest revise, Line 11 could be revised to:*

 '······the probability of Coulomb stress triggering between seismogenic structures included in the Taiwan Earthquake Model.'
We followed the reviewer's comment and revised the text accordingly.

*Lines 64-68: What value is used for the effective coefficient of friction (μ') in the Coulomb stress modelling?*
We first assumed a fixed $\mu'$ of 0.4. To quantify deviation on determining multiple-rupture pairs, we further considered $\mu'$=0.2 and 0.5, the boundaries of its reasonable range determined from focal mechanisms in Taiwan. Considering the stress threshold of $\Delta$CFS$\geq$0.1 bar and a distance threshold of 5 km, the potential paired structures were identified (Table 6). The results suggest slight differences within the reasonable effective friction coefficient (lines 54-56, 259-267).

*Line 115: These scaling relationships between magnitude and rupture area are presumably from Wells and Coppersmith (1994)? If so, they should be cited as such (though also see major comment #1)*
We followed the reviewer's comment and cited the reference accordingly.

*Line 144: Replace 'integrating' with 'combining,' to avoid any connotations that you are actually performing an integration in these equations.*
We followed the reviewer's comment and revised the text accordingly.

*Lines 220-221 (and 335): When referring to the Kaikōura earthquake, reference should be made to Hamling et al (2017). This is the original reference to this event and written by authors who made the primary observations of this multi-fault earthquake.*
We followed the reviewer's comment and cited the reference accordingly.

*Line 258: I think there is a typo here for describing the numeric value if the Hukou and Hsinchu fault recurrence intervals as '4.4 and 5.3'?*
We have revised the text as "their recurrence intervals become 4.4 and 5.3 times, respectively, longer than the cases without considering multiple-structure ruptures"

(lines 212-213).

*Figures: Figure 1 presents only a generic case of Coulomb stress changes around a fault. I would recommend also including a figure to show an example of this stress modelling from faults in the TEM. Maybe using the example of faults that are described further in Section 3.1?*
This figure has been removed.

---

## Author Comment (AC2)

We greatly appreciate the reviewer's insightful comments and have revised our manuscript, nhess-2022-46, entitled, "Quantifying the probability and uncertainty of multiple-structure rupture and recurrence intervals in Taiwan," accordingly. Below, we have quoted the comments in italics and provided our detailed responses. All the changes are underlined in the revised manuscript.

*The authors apply an analytical procedure inspired in Chan et al. (2020), which stems from the notion that when two faults interact each one gives part of its slip rate to the multiple-rupture process, while retaining the remaining slip rate to its individual-rupture process. Starting from this basic notion – the "partitioned slip rates" of Chan et al. (2020) - the authors use Kanamori's (1977) definition of moment magnitude, the definition of seismic moment, Wells and Coppersmith's (1994) scaling relations of moment magnitude with rupture area and the Gutenberg-Richter relation to obtain return periods for multiple ruptures and for individual ruptures on each fault. Key to the authors' reasoning is their assumption that the slip rate of each fault is partitioned between individual and multiple ruptures according to a "partitioned rate" given by their equations 10 and 11, which include "the magnitude of the multiple-structure rupture" (line 135) and "the displacement of the multiple-rupture structure" (line 136). This terminology reflects the fact (not explained in the manuscript) that the catalog used in the study considers characteristic ruptures only. The expression for C ifeatures the b-value of the Gutenberg-Richter relation. The authors present the expression for C as a logical conclusion of the Gutenberg-Richter relation, although I was not able to follow that logic. I was able to trace the definition of the partitioned rate C to Chan et al. (2020), but there too it was introduced without an explanation.*

I replied this comment through three aspects, algorithm description, scaling relation, and assumption of characteristic ruptures, detailed below.

To clearly describe our algorithm for evaluating the recurrence interval of multiple-structure ruptures, we first introduced the slip rate partitioned to individual structure ruptures (equations 8 and 9), followed by the obtained partitioned rates (equations 10 and 11). By combining them, the slip rate partitioned to the multiple-structure rupture from the original structures could be obtained (described in lines 122-140).

Although the scaling relations proposed by Wells and Coppersmith (1994) have been questioned by many modern models, especially for large megathrusts, Wang et al. (2016[b]) concluded a similar maximal magnitude of each seismogenic structure estimated from the relations of Wells and Coppersmith (1994) and Yen and Ma (2011), obtained from regressions of the rupture parameters of the earthquakes mainly from the Taiwan orogenic belt. Besides, to validate the sensitivity of our procedure to scaling, we implemented alternative relationships proposed by Yen and Ma (2011). Based on this relation, recurrence intervals for each multiple-structure rupture pairs were evaluated (Table 5). Comparing these with those obtained by Wells and Coppersmith's relations, shorter recurrence intervals were obtained, especially for those with larger magnitude. These results can be attributed to a smaller average displacement obtained for a large event that led to a shorter recurrence interval for the multiple-structure rupture (based on equation 17). We provided more detailed descriptions in lines 214-223, 292-298.

*The estimate of the multiple-rupture displacement is formally correct, albeit highly convoluted. But the estimate of the multiple-rupture slip rate through a sum defies logic, in my view (why sum slip rates interesting separate faults?) Also, as pointed out above, each parcel relies on a coefficient that was not sufficiently explained.*

To clearly describe our algorithm for evaluating the recurrence interval of multiple-structure ruptures, we have modified the manuscript to first introduce the slip rate partitioned to individual structure ruptures (equations 8 and 9), followed by the obtained partitioned rates (equations 10 and 11). By combining them, the slip rate partitioned to the multiple-structure rupture from the original structures can be obtained (described in lines 122-140).

The estimate of the multiple-rupture slip rate through a sum is based on the assumption that the slip of an earthquake is equal to the cumulative slip during an interseismic period. Since the slip of a multiple-structure rupture is the result of contributions from different structures, we sum the slip rates contributed from the individual structures.

*In section 3.2 the authors enlarge their approach to include more than two faults in interaction, increasing the complexity while inheriting the obscurity from the previous section.*

Earthquakes could be attributed to multiple (more than three) structures, for example, the 2010 El Mayor-Cucapah, US, earthquake; the 2016 Mw7.8 Kaikōura, New Zealand, earthquake. The procedure we proposed in Section 3.2 could quantify the return period of these earthquakes.

*In sections 3.3 and 4, the authors discuss some implications of their analysis for seismic hazard. Around line 245, the authors conclude that the possibility of multiple-rupture earthquakes reduces the hazard at the shorter return periods while increasing it at longer return periods. In line 255, the authors observe that "structures that pair with several cases of multiple-structure ruptures might be difficult to rupture solely". These observations are so clearly at odds with empirical evidence – which points to single-*

*fault rupture as the dominant contributor to hazard – that they should be regarded as indicating flaws of the approach.*

The description mentioned here is based on the comparison between models with and without multiple-structure ruptures. That is, the return period of a seismogenic structure could be longer if a part of its coupling rate will contribute to the multiple-structure rupture. Note that based on our procedure, a shorter return period is expected for a rupture on one individual structure than for a multiple-structure rupture. For example, we obtained a return period of 6,640 and 11,953 years for the Hsinchu fault and multiple-structure rupture of the Hukou fault and Hsinchu fault, respectively.

*The authors base their approach on a simplified view of stress transfer between faults: they ignore dynamic effects, pore-fluid effects and – surprisingly in view of published evidence – restrict the range of stress transfer to 5km. Although the title promised a quantification of the uncertainties, very little is done to quantify the errors that derive from such simplifications. In line 263 the authors state that their approach is a physics-based one. Unfortunatelly, it seems to have strayed strongly from the geological reality of earthquake generation. The authors recognize, to their credit, that the "analysis could be further improved through better understanding seismogenic structures" (line 278). I would take this conclusion even further and say that the analysis needs to be reformulated starting with a better understanding of seismogenic processes. For example, exploring empirical evidence of the occurrence and characteristics of multiple-rupture earthquakes in the available databases, in order to be able to subject their model to a reality check.*

We followed the reviewer's comment and included some more discussion on various physics-based components, including effective coefficient of friction (Table 6, lines 259-267), rake angle rotation (Table 8, lines 282-285), stress threshold of ΔCFS (Table

3, lines 89-94, 268-272), and distance threshold (Tables 3 and 7, lines 89-94, 268-272, 273-281). Note that we explained our model without implementing a poroelastic assumption, since previous studies (e.g., Chan and Stain, 2009) concluded that the differences in their results were trivial for assuming reasonable values of Skempton's coefficients (in between 0.5 and 0.9) and dry friction (0.75). Our approach indicated various rupture pairs and quantified uncertainties. These outcomes could be incorporated into a probabilistic seismic hazard assessment through a logic tree.

*In the present stage of development, I regret to conclude that I don't consider this research ready for publication.*

We appreciate the reviewer's very helpful comments. We hope the adjustments we have made accordingly to the manuscript meet the standards of *Natural Hazards and Earth System Sciences* and have made the manuscript to now ready for publication.

---

## Author Comment (AC3)

**Response to Reviewer #3 Anonymous Referee**

We highly appreciate the reviewer's insightful comments and have revised our manuscript, nhess-2022-46, entitled, "Quantifying the probability and uncertainty of multiple-structure rupture and recurrence intervals in Taiwan," accordingly. Below, we have quoted the comments in italics and provided our detailed responses. All the changes are underlined in the revised manuscript.

*Huang et al., in the manuscript "Quantifying the probability and uncertainty of multiple-structure rupture and recurrence intervals in Taiwan" presents a new approach by integrating the physics-based model (static Coulomb stress change) and statistic model (Gutenberg-Richter law) to evaluate the earthquake recurrence time for the possible multiple-rupture scenario. According to their assumption, multi-rupture only occurs if the stress transfer on the nearby fault reaches a certain value, and the slip rate of the multiple-rupture structure is the sum of the associated slip rate in related ruptures. Although I acknowledge this topic as a valuable contribution in the field of hazard assessment, however, this current manuscript needs improvements, especially I am still not clear about how the author partitioned the slip rates between different ruptures.*

To clearly describe our algorithm for slip rate partitioning, we revised our procedure to first introduce the slip rate partitioned to individual structure ruptures (equations 8 and 9), followed by the obtained partitioned rates (equations 10 and 11). By combining them, the slip rate partitioned to the multiple-structure rupture from the original structures could be obtained (described in Lines 123-140).

We hope the present version of the manuscript meets the standards of *Natural Hazards and Earth System Sciences* and is now ready for publication.

*The structure of the description.*

*I think section 3 is the core of the methodology in this study, as far as I can tell this study use simple equations, but the description makes it extremely difficult to follow. In general, I think the whole section of 3.1 and 3.2 should be reformulate, for example:*

*Equation (2),D^dot represents the slip rate, dose this slip rate indicates the long-term slip rate obtained from other measurements?*

The slip rate ($\dot{D}_{L1}$, shown in equation 2) is obtained from the TEM seismogenic structure database (Table 1).

To clearly describe our algorithm for the recurrence interval of multiple-structure ruptures, especially for slip rate partitioning, we modified Section 3 and hope the current version achieves the desired clarity (lines 97-190).

*Equation (7), the author used the Mw-Mo scaling law by Kanamori (1977), but the equation in the manuscript is from Hanks and Kanamori (1979) with the unit of dyne-cm.*

We thank the reviewer highly for having identified this oversight in our paper. We have revised the manuscript accordingly and simplified equation 7.

*Equation (8) and (9), there appear two parameters D_L1' and D_L2' with no explanations until equation (12) and equation (13).*

*Equation (10), dose the ML1 indicates the maximum magnitude in L1 ? D_L1+L2 is the displacement of the multiple-structure rupture, dose this means D_L1+L2 = D_L1 + D_L2? More practical parameter annotation should be carefully addressed.*

*Equation (14), this equation is hard to follow, in Line 146 : the sum of the slip rates for the multiple-structure…. I don't understand what is the sum of the slip rates for the multiple-structure? and this statement is not correspond to the equation (14).*

To clearly describe our algorithm for evaluating the recurrence interval of multiple-structure ruptures, we first introduced the slip rate partitioned to individual structure ruptures (equations 8 and 9), followed by the obtained partitioned rates (equations 10 and 11). By combining them, the slip rate partitioned to the multiple-structure rupture from the original structures could be obtained (described in lines 123-140).

*The author took 1906 Meishan earthquake as an example, they argued that closed-by Chiayi frontal structure also ruptured during the coseismic period because liquefaction took place on the west of the Meishan fault, however, I think this statement is little-bit weak because liquefaction could occur when the stress is perturbated through seismic wave propagation from the mainshock.*

To better illustrate the rupture behavior of the Maishan earthquake, we provided evidence such as the larger magnitude than the characteristic magnitude of the Meishan fault, the focal mechanism of oblique thrust faulting being oriented in the northeast–southwest direction, and the large ground shaking with liquefaction that took place to the west. All infer the Chiayi frontal structure might rupture simultaneously.

*Also, I got confused when reading the line from 286 to 288, dose the author really hints that Meishan earthquake is initiated on the Chiayi frontal structure?*

The description of the simplified Coulomb stress change model has been removed.

*For model uncertainty, this sensitivity test is focus only on the rake angles for estimating the Coulomb stress change, I was wondering what if they change the friction coefficient? Friction coefficient also plays an important role on evaluating the stress impart from the mainshock, especially recent studies suggest that friction coefficient is depth dependence (i.e., Carpenter et al., 2012,2015). Besides the Coulomb stress model,*

*G-R law also make a strong contribution on this approach, I am wondering if they consider different type of G-R law will change the result significantly (for example the truncated model)?*

We followed the reviewer's comment and discussed the impact of the friction coefficient. We considered $\mu'=0.2$ and 0.5, the boundaries of its reasonable range determined from focal mechanisms in Taiwan. Considering the stress threshold of $\Delta CFS \geq 0.1$ bar and a distance threshold of 5 km, the potential paired structures were identified (Table 6). The results suggest slight differences within the reasonable effective friction coefficient (lines 54-56, 259-267). Besides, we explained our model without implementing a poroelastic assumption since previous studies (e.g., Chan and Stain, 2009) concluded that the differences in their results were trivial for assuming reasonable values of Skempton's coefficients (between 0.5 and 0.9) and dry friction (0.75) (lines 259-262).

*minor comments:*

*Line 131, show in equation 1 -> equation 3*

This sentence has been removed.

*Line 157, what is characteristic earthquake means? rupture or slip or magnitude?*

This paragraph has been removed.

*Line 159~ , The author addresses the exact value of each parameter very carefully, but I do think those repetitive equations and number should be removed and only use a simple table to present. Line 284, missing the ID for Chiayi frontal structure.*

This paragraph has been removed.

---

## Author Comment (AC4)

We highly appreciate the reviewer's insightful comments and have revised our manuscript, nhess-2022-46, entitled, "Quantifying the probability and uncertainty of multiple-structure rupture and recurrence intervals in Taiwan," accordingly. Below, we have quoted the comments in italics and provided our
5  detailed responses. All the changes are underlined in the revised manuscript.

*Nevertheless, theories or new approaches and assumptions should rely on physical processes and need to incorporate the comprehension of the reality and I have some doubts about validity of some assumptions, namely:*

10 *- what is the meaning of summing slip rates of the different faults?*

The estimate of the multiple-rupture slip rate through a sum is based on the assumption that the slip of an earthquake is equal to the cumulative slip during an interseismic period. Since the slip of a multiple-structure rupture is the result of contributions from different structures, we sum the slip rates contributed from the individual structures.

15  To better describe our algorithm for evaluating the recurrence interval of multiple-structure ruptures, we have adjusted the manuscript to first introduce the slip rate partitioned to individual structure ruptures (equations 8 and 9), followed by the obtained partitioned rates (equations 10 and 11). By combining them, the slip rate partitioned to the multiple-structure rupture from the original structures could be obtained (described in lines 123-140).

20

*- Why the distance between two faults must be less than 5 km? is there any evidence that there is no Coulomb stress transfer for distances greater than 5 km that can trigger a fault? I could recommend a little more discussion on this issue.*

We expected that a long distance between two structures could result in it being difficul for the pair to
25  rupture simultaneously. Thus, we followed the criterion by the UCERF3 (Field et al., 2015) and assumed a distance threshold of 5 km.

We were aware that an earthquake with a large coseismic slip dislocation could result in a significant stress change at distance and then searched the pairs with longer distances and significant stress increase.

Two additional distance thresholds of 10 and 20 km were considered (Table 7). Generally, potential magnitudes of these structures are relatively large, which could result in leger stress perturbation.

*The title "Quantifying the probability and uncertainty ...." do not reflect the content of the paper, in my opinion, as it leads to an expectation of a sensitivity study on key parameters that might have impact on results. The only parameters changed were the Coulomb stress and the structure rake angle. Are there any other parameters that can affect results? Were these parameters chosen because they are the ones with the most impact? I was expecting a more exhaustive study on that.*

We followed the reviewer's comment and included additional discussion on uncertainties from various parameters, including the effective coefficient of friction (Table 6, lines 259-267), rake angle rotation (Table 8, lines 282-285), stress threshold of ΔCFS (Table 3, lines 89-94, 268-272), and distance threshold (Tables 3 and 7, lines 89-94, 268-272, 273-281). Our approach indicated various rupture pairs and quantified uncertainties. These outcomes were able to be incorporated into a probabilistic seismic hazard assessment through a logic tree.

*Finally, I would suggest a different way to present so many and so similar equations, as it become difficult and not very interesting to follow.*

We have rearranged the description of the procedure, simplified some equations, and removed several examples in Chapter 3. Hopefully, the current manuscript is easily understandable and meets the standards of *Natural Hazards and Earth System Sciences*.

---

## Author Comment (AC5)

**Quantifying the probability and uncertainty of multiple-structure rupture and recurrence intervals in Taiwan**

Chieh-Chen Chang1, Chih-Yu Chang1, Chung-Han Chan1,2

1Department of Earth Sciences, National Central University, Taoyuan, 32001, Taiwan

5 2Earthquake-Disaster & Risk Evaluation and Management (E-DREaM) Center, National Central University, Taoyuan, 32001, Taiwan

Correspondence to: Chung-Han Chan (hantijun@googlemail.com)

Abstract. This study identifies structure pairs with the potential for simultaneous rupture in a coseismic period via Coulomb stress change and quantifies their rupture recurrence intervals and uncertainties according to the Gutenberg-Richter law and

- 10 the empirical formula of rupture parameters. To assess the potential for a multiple-structure rupture, we calculated the probability of Coulomb stress triggering between seismogenic structures included in the Taiwan Earthquake Model. We assumed that a multiple-structure rupture would occur if two structures could trigger each other by enhancing the plane with thresholds of a Coulomb stress increase and the distance between the structures. According to different thresholds, we identified various sets of seismogenic structure pairs. To estimate the recurrence intervals for multiple-structure ruptures, we
- 15 implemented a scaling law and the Gutenberg-Richter law in which the slip rate could be partitioned based on the magnitudes of the individual structure and multiple-structure ruptures. In addition, considering that a single structure may be involved in multiple cases of multiple-structure ruptures, we developed new formulas for slip partitioning in a complex fault system. By implementing the range of slip area and slip rate of each structure, the magnitudes and recurrence intervals of multiple-structure ruptures could be estimated. Due to a larger characteristic magnitude and a larger displacement of the multiple-structure rupture,
- 20 the rupture's recurrence interval could be longer. Therefore, application of the multiple-structure rupture could lead to an increase in seismic hazard in a long return period, which would be crucial for the safety evaluation of infrastructures, such as nuclear power plants and dams.

**1** Introduction**

A rupture taking place along several fault segments and/or structures can cause an earthquake with a large magnitude (e.g.,

25 Yen and Ma, 2011) and often leads to disaster. The 1935 ML7.1 Hsinchu-Taichung, Taiwan, earthquake is an example. This event is attributed to a rupture on the Shihtan and Tunzijiao faults and resulted in more than 3,000 fatalities and the destruction of more than 60,000 buildings. According to the fault parameters determined by Shyu et al. (2020), either the Shihtan or Tunzijiao fault could cause an earthquake with a maximum magnitude of only 6.6 (Wang et al., 2016a). This case raises the importance of multiple-structure ruptures on seismic hazard assessment.

[revised manuscript text omitted]

We have identified potential structures that might rupture in a coseismic period. To understand the activities of these multiplestructure rupture cases, we will next propose a procedure to evaluate their recurrence intervals.

**3** Recurrence interval of the multiple-structure rupture**

The recurrence interval is a critical parameter in probabilistic seismic hazard analysis. Here, we are going to calculate the recurrence interval of multiple-structure ruptures and discuss their impact on seismic hazards.

**3.1 Recurrence interval of multiple-structure ruptures**

100 According to the TEM seismogenic structure database (Shyu et al., 2020) and the TEM PSHA2020 (Chan et al., 2020), the rupture recurrence interval (denoted as  $R_{L1}$ ) of a single seismogenic structure (L1),  $R_{L1}$ , can be evaluated as the ratio of slip of a characteristic earthquake to slip rate (denoted as  $D_{L1}$  and  $\dot{D}_{L1}$ , respectively):

$$R_{L1} = \frac{D_{L1}}{\dot{D}_{L1}}.$$
 (2)

To evaluate the seismic rate of a multiple-structure rupture on two seismogenic structures (L1 and L2), we implemented the 105 Gutenberg-Richter law to describe the relationship between earthquake frequency N and magnitude M:

$$\log(\mathbf{N}) = a - bM. \tag{3}$$

Considering the different moment magnitudes between single-structure and multiple-structure ruptures, the ratio of earthquake frequency to slip-rate partitioning could be evaluated. The moment magnitude  $(M_w)$  of the multiple-structure rupture could be evaluated according to the rupture area (denoted as *A*) and fault types of the two seismogenic structures. In the TEM structure

110 database, determination of rupture magnitude (Table 1) is based on the scaling law proposed by Wells and Coppersmith (1994), represented as:

$$M_w = 4.33 + 0.90 \times \log(A) \dots \text{ for reverse faulting;}$$
(4)

$$M_w = 3.98 + 1.02 \times \log(A) \dots \text{ for strike-slip faulting;}$$
(5)

- $M_w = 3.93 + 1.02 \times \log(A) \dots \text{ for normal faulting.}$ (6)
- 115 We first follow the procedure of the TEM model to implement these scaling relations and then evaluate uncertainty of this procedure considering different scaling relations.

Based on the Mw-M0 scale (Kanamori, 1977) and the definition of seismic moment, average displacement of a seismogenic structure (*D*, in meters) could be evaluated according to  $M_w$  and *A* (in km2):

$$\underline{D} = \frac{10^{\frac{2}{3}M_{W}} \times 10^{-15.85}}{3A}.$$
(7)

**120 Here we first implement the same scaling relations as those for the TEM model and then evaluate uncertainty of this procedure considering different scaling relations.**

[revised manuscript text omitted]

165
$$R_{L1} = \frac{D_{L1}}{D_{L1}'}$$
 (21)

and

$$R_{Lx} = \frac{D_{Lx}}{\dot{D}_{Lx}}$$
, respectively.

A single earthquake could be attributed to multiple (more than three) structures, for example, the 2010 El Mayor-Cucapah, US, earthquake (Wei et al., 2011); the 2016 Mw7.8 Kaikōura, New Zealand, earthquake (Hamling et al., 2017). In such special

(22)

170 cases, the recurrence interval can be also evaluated through the procedure mentioned above. For example, the Chiayi frontal structure (ID 21, here denoted as *L*1) could trigger the Meishan fault (ID 20, here denoted as *L*2) and the Tainan frontal structure (ID 41, here denoted as *L*3), respectively, in some criteria (Table 3), inferring the possibility of multiple ruptures in an event. We assumed this event is reverse faulting and evaluated its fault area and moment magnitude accordingly, described in the following:

175  $A_{L1+L2+L3} = 371.7 + 1580.88 + 1722.64 = 3675.22 \ km^2$ ;

$$M_{W_{L1+L2+L3}} = 4.33 + 0.90 \times log (3675.22) = 7.54;$$

$$D_{L1+L2+L3} = \frac{10^{(7.54+10.73)\times\frac{2}{3}\times10^{-12}}}{3\times10^{11}\times3675.22} = 2.305 \, m;$$

$$\dot{D}_{L1}' = \frac{1580.88 \times 3.36 \times 1.71}{(1580.88 \times 1.71) + (1952.58 \times 1.829 \times 10^{1.1 \times (7.21 - 7.29)}) + (3303.52 \times 2.233 \times 10^{1.1 \times (7.21 - 7.5)}) + (3675.22 \times 2.305 \times 10^{1.1 \times (7.21 - 7.54)})} = 0.708 \ mm/s$$
year;

180
$$\dot{D}_{L1+L2+L3}^{L1} = \frac{1580.88 \times 3.36 \times 2.305 \times 10^{1.1 \times (7.21-7.54)}}{(1580.88 \times 1.71) + (1952.58 \times 1.829 \times 10^{1.1 \times (7.21-7.29)}) + (3303.52 \times 2.233 \times 10^{1.1 \times (7.21-7.5)}) + (3675.22 \times 2.305 \times 10^{1.1 \times (7.21-7.54)})} = 0.414 \ mm/year;$$

$$\dot{D}_{L1+L2+L3}^{L2} = \frac{371.7 \times 2.51 \times 2.305 \times 10^{1.1 \times (6.6-7.54)}}{(371.7 \times 0.89) + (1952.58 \times 1.829 \times 10^{1.1 \times (6.6-7.29)}) + (3675.22 \times 2.305 \times 10^{1.1 \times (6.6-7.54)})} = 0.114 \text{ mm/year; and}$$
$$\dot{D}_{L1+L2+L3}^{L3} = \frac{1722.64 \times 0.92 \times 2.305 \times 10^{1.1 \times (7.24-7.54)}}{(1722.64 \times 1.74) + (3303.52 \times 2.233 \times 10^{1.1 \times (7.24-7.5)}) + (3675.22 \times 2.305 \times 10^{1.1 \times (7.24-7.54)})} = 0.159 \text{ mm/year.}$$

Note that L2 and L3 will not rupture together:

185  $\dot{D}_{L1+L2+L3} = 0.414 + 0.114 + 0.159 = 0.687 \, mm/year;$

$$R_{L1} = \frac{1.71}{0.708} \times 1000 = 2415 \text{ years};$$
 then
 $R_{L1+L2+L3} = \frac{2.305}{0.687} \times 1000 = 3355 \text{ years}.$

Thus, rupture probability of multiple structures could be quantified, which could constrain subsequent probabilistic seismic hazard assessment.

**190 3.3 Multiple-structure rupture recurrence intervals and uncertainties**

According to the structure parameters (Table 1), the recurrence intervals of each pair of potential multiple-structure ruptures can be evaluated (Table 2). Here, we consider the 17 pairs with  $\Delta CFS \ge 0.1$  bar and distance  $\le 5.0$  km and evaluated their potential magnitudes and recurrence intervals by implementing the range of slip area and slip rate of each structure (Table 1). Considering epistemic uncertainties, the largest magnitude is expected if the maximum slip areas of the two structures are

195 assumed (based on equations 4-6). Also, the shortest recurrence interval is expected if the minimum slip area and maximum slip rate are assumed (based on equations 4-17).

In comparison with the recurrence intervals of the original structures without considering a multiple-structure rupture (Table 1), longer recurrence intervals are expected for multiple-structure ruptures and individual structures due to slip partitioning. For example, the recurrence interval of the Chiayi frontal structure (ID 21) has been extended from 510 to 1,724 years. Based

200 on these results, the seismic hazard level for a short return period (e.g., 475 years, corresponding to a 10% probability in 50 years) would be lower.

Additionally, our results show that a single seismogenic structure sometimes pairs with several cases of multiple-structure ruptures. For example, the Hukou fault (ID 4) potentially ruptures with the Shuanglianpo structure (ID 2), the Fengshan river strike-slip structure (ID 5), and the Hsinchu fault (ID 6), while the Hsinchu fault (ID 6) could also result in multiple-segment

- 205 ruptures with the Hsinchu frontal structure (ID 8) and the Touhuanping structure (ID 9). Besides these two cases associated with three rupture pairs, several structures could be associated with two multiple-structure pairs (Table 2), raising the importance of implementing slip partitioning from a single structure to several multiple-structure ruptures. Based on our analysis, it might be difficult for the structures that pair with several cases of multiple-structure ruptures might to rupture solely. That is, based on equations 18 to 22, the slip rate of these structures could be partitioned to several cases of multiple-structure
- 210 ruptures, resulting in longer recurrence intervals. For example, the Hukou fault (ID 4) and the Hsinchu fault (ID 6) involved four and three pairs of multiple-structure ruptures, respectively (Table 2), and their recurrence intervals became 4.4 and 5.3 times, respectively, longer than the cases without considering multiple-structure ruptures (Table 4).

Our calculations of recurrence interval for the multiple-structure ruptures are based on the scaling relations proposed by Wells and Coppersmith (1994). These relationships were obtained based on the global data summarized decades ago. To validate the

215 sensitivity of our procedure to scaling, here we implement alternative relationships proposed by Yen and Ma (2011), who investigated the rupture parameters of the earthquakes mainly from the Taiwan orogenic belt. This relation illustrates average displacement of a seismogenic structure (*D*, in meters) as a constant:

Log(D) = -0.32.

(23)

Based on this relation, recurrence intervals for each multiple-structure rupture pairs were evaluated (Table 5). Comparing these to those obtained by Wells and Coppersmith's relations, shorter recurrence intervals were obtained, especially for those with larger magnitude. These results can be attributed to a smaller average displacement obtained for a large event that led to a shorter recurrence interval for the multiple-structure rupture (based on equation 17).

**4 Discussion and conclusion**

**4.1 Interaction between structures and possible coseismic ruptures**

225 In this study, we explored possible coseismic multiple-structure ruptures and quantified their recurrence intervals by implementing the Coulomb stress change and the Gutenberg-Richter law, respectively. The analyzing procedure we proposed is based on physics- and statistics-based models, and the outcomes are reproducible.

We compared our results with Shyu et al.'s (2020) conclusion that some seismogenic structure pairs—such as the Hsinchu fault (ID 6) and the Hsinchu frontal structure (ID 8), the Touhuanping fault (ID 9) and the Miaoli frontal structure (ID 10), the

230 Meishan fault (ID 20) and the Chiayi frontal structure (ID 21), and the Chiayi frontal structure (ID 21) and the Tainan frontal structure (ID 41)—could rupture simultaneously. Their findings were consistent with our results based on the Coulomb stress triggering.

Additionally, Shyu et al. (2020) suggested some other structure pairs for multiple-structure ruptures, such as the Shihtan fault (ID 13) and Tuntzuchiao fault (ID 15), the Houchiali fault (ID 25) and the Tainan frontal structure (ID 41), and the Chaochou

- fault (ID 29) and the Hengchun fault (ID 30). These pairs, however, do not fit our hypothesis. Take the Shihtan and Tuntzuchiao faults, for example. The rupture of the Tuntzuchiao fault could result in a Coulomb stress increase of more than 0.1 bar in 79% of the sub-faults of the Shihtan fault, whereas only 2% of the sub-fault in the Tuntzuchiao fault would be triggered when the Shihtan fault dislocates (Table S1). Note that the 1935 Hsinchu-Taichung earthquake is attributed to a coseismic rupture on the two faults. Previous studies (Yan, 2016; Su, 2019) indicated that this earthquake did not initiate on either the Shihtan or
- 240 the Tuntzuchiao fault, but on a blind fault linking the two. The database we accessed (Shyu et al., 2020) did not include this blind structure. Our analysis could be further improved through better understanding seismogenic structures. In addition, we discussed the interaction between structures through a kinematic model; it is desired to further incorporate dynamic models (e.g., Brodsky and van der Elst 2014; Jiao et al., 2022; Lin, 2021; Ulrich et al 2018) to constrain the behaviors of multiple-structure ruptures.
- In 1906, an earthquake with magnitude 7.1 occurred due to the rupture of the Meishan fault (ID 20). Considering its fault geometry, the characteristic magnitude of this fault is only 6.6; therefore, this event with a larger magnitude could be associated with a multiple-structure rupture. In addition, the focal mechanism of this earthquake suggests that this event cannot be attributed solely to the rupture on the Meishan fault. The first motions of P- and S-waves recorded by the seismograph suggest oblique thrust faulting oriented in the northeast-southwest direction, with a small right-lateral component (Liao et al., 2018).
- 250 Besides, large ground shaking with liquefaction took place to the west of the Meishan fault during the coseismic period (Omori, 1906). Thus, the Chiayi frontal structure might rupture simultaneously. Considering parameters of the Meishan fault and the

Chiayi frontal thrust (structure geometry, characteristic slip), when the Meishan fault is dislocated, the Coulomb stress on 64% of the Chiayi frontal structure plane may rise by more than 0.1 bar, and when the Chiayi frontal structure is dislocated, 72% of the Meishan fault could be closer to failure (Table S1). In addition, the distance between the two faults is 1.87 km (Table

S2). Therefore, we concluded that these two structures could have mutually ruptured in a coseismic period and resulted in an event with magnitude 7.1 in 1906.

**4.2 Uncertainty of the Coulomb stress model and recurrence interval**

In this study, we identified potential rupture pairs by considering Coulomb stress change along the shear and normal components and the effective friction coefficient (equation 1). We simplified this model without implementing a poroelastic

- 260 assumption (Beeler et al., 2000), since previous studies (e.g., Chan and Stain, 2009) concluded that the differences in their results were trivial for assuming reasonable values of Skempton's coefficients (between 0.5 and 0.9) and dry friction (0.75). The effective friction coefficient ( $\mu$ ') could alter the impact of normal stress change on the Coulomb stress change ( $\Delta CFS$ ). To quantify the deviation on determining multiple-rupture pairs, we further considered  $\mu$ '=0.2 and 0.5, the boundaries of its reasonable range determined from focal mechanisms in Taiwan (Hsu et al., 2010). Considering the stress threshold of
- 265  $\Delta CFS \ge 0.1$  bar and distance threshold of 5 km, the potential paired structures were identified (Table 6). The results suggest slight differences in the reasonable effective friction coefficient in between 0.2 and 0.5.

In this study, we identified potential rupture pairs by considering thresholds of stress change and structure distance. We implemented four threshold sets of Coulomb stress change (+0.01, +0.05, +0.1, and +0.2 bars) and two for distance between structures (2.5 and 5.0 km) to identify plausible pairs for multiple-structure rupture (Table 3). Also, the uncertainty of the

270 structure rake angle could result in deviation. Our standard procedure assumed a fixed rake angle of each structure according to its rupture type (Table 1), while in reality its rupture orientation could alter slightly in small patches of the structure plane.

We expected a long distance between two structures could make it difficult for the two structures to rupture simultaneously. Thus, we followed the criterion by the UCERF3 (Field et al., 2015) and assumed a distance threshold of 5 km. We are aware that an earthquake with a large coseismic slip dislocation could result in significant stress change in far field and then search

- the pairs with longer distances and significant stress increase. Two additional distance thresholds of 10 and 20 km were considered (Table 7), and 6 and 9 additional pairs that might rupture in a coseismic period were identified, respectively. Generally, potential magnitudes of these structures are relatively large, which could result in larger stress perturbation. For example, the Chiayi frontal structure could cause an event with magnitude 7.21, resulting in a Coulomb stress increase of more than 0.1 bar in 91% of the sub-faults of the Chungchou structures, when 80% of the sub-fault in the Chiayi frontal structure 280 would be triggered when the Chungchou structures dislocates with an M6.89 event (Table S1).
  - To evaluate the impact of rake angle orientation, we evaluated the Coulomb stress change on the receiving structure with different rotated rake angles (i.e.,  $\pm 10^{\circ}$  and  $\pm 20^{\circ}$ ). The results showed that the larger the rotated rake angles implemented for

the receiver structures, the fewer structure pairs were identified (Table 8). Note that 11 pairs were identified even when the rakes rotated for  $\pm 20^{\circ}$ , suggesting their robustness for coseismic multiple-structure rupture.

- 285 Besides the uncertainty of structure pair identification, uncertainties in the rupture parameters of the multiple structures could be evaluated. Considering the range of the structures' slip areas (Table 1), magnitude intervals of multiple-structure ruptures could be estimated (Table 2). That is that the largest magnitude for multiple-structure rupture can be obtained when we consider the maximum slip areas of the two structures (based on equations 4-6). By further implementing structure slip rates, recurrence intervals can be quantified: the minimum slip area and maximum slip rate obtains the shortest recurrence interval (based on
- 290 equations 4-17).

Rupture recurrence intervals could also be influenced by the implemented scaling relations. We proposed two relations, that is, in addition to the well-known relations by Wells and Coppersmith (1994), we also used the relations proposed by Yen and Ma (2011) that were obtained from the observations mainly from Taiwan. Since the local relationships (Yen and Ma, 2011) infer a smaller displacement, shorter recurrence intervals were obtained (Table 5). Besides, although the scaling relations

295 proposed by Wells and Coppersmith (1994) have been questioned by many modern models, especially for large megathrusts (e.g., Stirling et al., 2013), Wang et al. (2016b) concluded a similar maximal magnitude of each seismogenic structure estimated from the relations of Wells and Coppersmith (1994) and Yen and Ma (2011).

For recurrence interval, the magnitude-frequency distribution on a single-structure plays an important role. Evaluating the rupture recurrence interval on a single structure could be based on various models, for example, the Gutenberg-Richter law

- 300 (Gutenberg and Richter, 1944), the characteristic earthquake model (Youngs and Coppersmith 1984; Hecker et al 2013; Stirling and Zungia 2017) in addition to others (e.g., Geist and Parsons 2019; Page et al 2021). In this study, we evaluated the rupture recurrence interval as the ratio of slip of a characteristic earthquake (with maximum magnitude of the structure) and slip rate, shown as equation (2), based on the assumption proposed by the TEM seismogenic structure database (Shyu et al., 2020) and the TEM PSHA2020 (Chan et al., 2020). This factor could be replaced by other magnitude-frequency distributions since the
- 305 recurrence interval of the multiple-structure rupture in our procedure is based on slip rate partitioned from individual structure ruptures (shown as equations 8-9, 14, 18, and 20).

Based on our analyses mentioned above, deviations of multiple-structure rupture pairs were indicated, and epistemic uncertainties of corresponding parameters were quantified, providing a better understanding of multiple-structure rupture behaviors, beneficial to subsequent research, such as the probabilistic seismic hazard assessment (PSHA), mentioned below.

**310 4.3 Application of multiple-structure rupture to probabilistic seismic hazard analysis**

Conducting a PSHA requires understanding the recurrence interval and potential magnitude of each seismogenic source, and implementing a hazard model with multiple-structure rupture could improve the assessment. Take the PSHA proposed by the TEM in 2020 (TEM PSHA2020, Chan et al., 2020) as an example—considering the cases of multiple-structure ruptures, the

hazard levels in the regions close to the Chaochou fault (ID 29) and the Tainan frontal structure (ID 41) increased significantly

315 for a long return period (recurrence interval of 2,475 years, see Fig. 3 of Chan et al., 2020). Chan et al.'s study (2020) indicated that the seismic hazard level would be misestimated if the probability of multiple-structure rupture is not implemented.

Seismic hazard analysis plays an essential role in constructing infrastructures, such as nuclear power plants, that require assuming a long return period. Thus, a seismogenic source with a long recurrence interval could be crucial for the analysis, raising the importance of multiple-fault rupture with a larger magnitude (larger than the characteristic earthquake of each structure).

The possibility of multiple-structure rupture used to be determined based on geological and geomorphological evidence with subjective judgments. Our study implemented a Coulomb stress change combined with statistical approaches to indicate

In addition, our approach indicated various rupture pairs and quantified uncertainties. These outcomes could be incorporated
into a PSHA through a logic tree. For example, larger weightings (possibilities) could be assumed for the pairs that fulfill more thresholds in the distance, Coulomb stress change (Table 3) and rotated rake angles (Table 8). That includes, for instance, the Shuanglianpo fault (ID 2) and the Hukou fault (ID 4); the Hukou fault (ID 4) and the Fengshan River strike-slip structure (ID 5); the Hsinchu fault (ID 6) and the Hsinchu frontal structure (ID 8); the Miaoli frontal structure (ID 10) and Tuntzuchiao fault (ID 15); the Muchiliao-Liuchia fault (ID 22) and the Chungchou structure (ID 23); and the Chishan fault (ID 26) and the 330 Fengshan structure (ID 45).

**4.4 Multiple structure rupture (with more than three structures)**

multiple-structure rupture pairs, which is transparent and reproducible.

320

The 2016  $M_w7.8$  Kaikōura, New Zealand, earthquake is an event resulting from ruptures on multiple structures. Hamling et al. (2017) indicated that this earthquake included ruptures along four major faults and up to 12 minor faults. From this case, we are aware that multiple-structure rupture is not limited to the combination of two seismogenic structures.

- Based on the multiple-structure rupture database proposed in this study (Table 2), several structures are associated with several possible rupture pairs. For instance, the Shuanglianpo fault (ID 2) may cause coseismic rupture with the Yangmei structure (ID 3) and the Hukou fault (ID 4), and the Hukou fault (ID 4) may link with the Fengshan River strike-slip structure (ID 5) and the Hsinchu fault (ID 6). Since our approach is based on a static Coulomb stress change, it is difficult to evaluate the temporal evolution of rupture probability. The possibility of a multiple-structure rupture in a coseismic period might be
- 340 overestimated. One potential solution is to implement a dynamic model (e.g., a discrete element model; Cundall and Strack, 1979) that simulates temporal distribution of displacement and stress fields and could be helpful in identifying plausible structures that perhaps rupture within a coseismic period.

**5** Acknowledgements**

This study was supported by the Ministry of Science and Technology in Taiwan under the grants MOST 109-2116-M-008 -

345 029 -MY3, MOST 110-2124-M-002 -008, and MOST 110-2634-F-008-008. This work is financially supported by the Earthquake-Disaster & Risk Evaluation and Management Center (E-DREaM) from the Featured Areas Research Center Program within the framework of the Higher Education Sprout Project by the Ministry of Education in Taiwan. The authors would like to thank Jack Williams, João Fonseca, Alexandra Carvalho and an anonymous reviewer for their constructive comments.

[revised manuscript text omitted]